# A *qnr*-plasmid allows aminoglycosides to induce SOS in *Escherichia coli*

Anamaria Babosan[1], David Skurnik[2], Anaëlle Muggeo[3], Gerald B Pier[4], Zeynep Baharoglu[5], Thomas Jové[6], Marie-Cécile Ploy[6], Sophie Griveau[7], Fethi Bedioui[7], Sébastien Vergnolle[8], Sophie Moussalih[1], Christophe de Champs[3], Didier Mazel[5], Thomas Guillard[3]*

[1]Inserm UMR-S 1250 P3Cell, SFR CAP-Santé, Université de Reims-Champagne-Ardenne, Reims, France; [2]Assistance Publique-Hôpitaux de Paris, Department of Clinical Microbiology, Necker-Enfants Malades University Hospital, Université de Paris, 75015 Paris, France. INSERM U1151-Equipe 1, Institut Necker-Enfants Malades, Université de Paris, 75015 Paris, France. Division of Infectious Diseases, Brigham and Women's Hospital, Harvard Medical School, Boston, MA 02115, USA., Boston, United States; [3]Inserm UMR-S 1250 P3Cell, SFR CAP-Santé, Université de Reims-Champagne-Ardenne, Reims, France. Laboratoire de Bactériologie-Virologie-Hygiène Hospitalière-Parasitologie- Mycologie, CHU Reims, Hôpital Robert Debré, Reims, France; [4]Division of Infectious Diseases, Department of Medicine, Brigham and Women's Hospital, Harvard Medical School, Boston, United States; [5]Institut Pasteur, Unité Plasticité du Génome Bactérien, CNRS UMR3525, Paris, France; [6]Université de Limoges, Inserm, CHU Limoges, RESINFIT, UMR 1092, Limoges, France; [7]Chimie ParisTech, PSL Research University, CNRS, Institute of Chemistry for Life and Health Sciences, Paris, France; [8]Laboratoire d'Hématologie, CH de Troyes, Troyes, France

**\*For correspondence:**
tguillard@chu-reims.fr

**Competing interest:** The authors declare that no competing interests exist.

**Abstract** The plasmid-mediated quinolone resistance (PMQR) genes have been shown to promote high-level bacterial resistance to fluoroquinolone antibiotics, potentially leading to clinical treatment failures. In *Escherichia coli*, sub-minimum inhibitory concentrations (sub-MICs) of the widely used fluoroquinolones are known to induce the SOS response. Interestingly, the expression of several PMQR *qnr* genes is controlled by the SOS master regulator, LexA. During the characterization of a small *qnrD*-plasmid carried in *E. coli,* we observed that the aminoglycosides become able to induce the SOS response in this species, thus leading to the elevated transcription of *qnrD*. Our findings show that the induction of the SOS response is due to nitric oxide (NO) accumulation in the presence of sub-MIC of aminoglycosides. We demonstrated that the NO accumulation is driven by two plasmid genes, ORF3 and ORF4, whose products act at two levels. ORF3 encodes a putative flavin adenine dinucleotide (*FAD*)-binding oxidoreductase which helps NO synthesis, while ORF4 codes for a putative fumarate and nitrate reductase (*FNR*)-type transcription factor, related to an $O_2$-responsive regulator of *hmp* expression, able to repress the Hmp-mediated NO detoxification pathway of *E. coli*. Thus, this discovery, that other major classes of antibiotics may induce the SOS response could have worthwhile implications for antibiotic stewardship efforts in preventing the emergence of resistance.

## Editor's evaluation

This manuscript describes how a small plasmid containing a quinolone resistance determinant changes the cellular response to sub-inhibitory concentrations of Tobramycin. The authors report that *E. coli* cells carrying this plasmid undergo nitrosative stress mediated by two previously

uncharacterized genes, which results in induction of the SOS response. These findings are interesting and relevant for readers across microbiology and genetics fields.

## Introduction

*Escherichia coli* is a well-known commensal of the gastrointestinal tract of vertebrates, including humans (*Tenaillon et al., 2010*), but several strains can also cause enteric and extra-enteric diseases such as urinary tract infection or sepsis (*Kaper et al., 2004*). *E. coli* is in the fluoroquinolone's spectrum of action, and these antibiotics are widely used to treat such infections (*Lode, 2014*; *Rice, 2012*). Historically, fluoroquinolone resistance was found to develop solely through chromosome-mediated mechanisms, but plasmid-mediated quinolone resistance (PMQR) genes are now being identified more and more frequently in clinical isolates (*Strahilevitz et al., 2009*). *qnr* genes are important PMQR determinants, with six families described so far (*qnrA*, *qnrB*, *qnrC*, *qnrD*, *qnrS*, and *qnrVC*) (*Ruiz, 2019*).

Among these *qnr* genes, *qnrD* was first described in *Salmonella enterica* isolates located on a 4270-bp long non-conjugative plasmid (p2007057), a very different context from that of the other *qnr* genes in terms of plasmid size (*Cavaco et al., 2009*). Soon after, we reported for the first time the presence of smaller *qnrD*-plasmids (~2.7 kb, with pDIJ09-518a as the archetype) in several bacteria belonging to *Morganellaceae* and proposed, what is now considered as the likely scenario, that the origin of *qnrD* lies within an as-yet-unidentified progenitor from this family (*Guillard et al., 2014*; *Ruiz, 2019*). pDIJ09-518a is a 2683-bp long plasmid-harbouring four open reading frames (ORFs), including *qnrD*, and exhibits only 53% identity with the plasmid found in *S. enterica* (*Guillard et al., 2012*; *Figure 1*). No function has yet been found for ORF2, ORF3, or ORF4. *qnrD*-plasmids can be roughly divided into two categories: the pDIJ09-518a-like plasmids and the p2007047-like plasmids. The ORF3 and ORF4 can only be found in the pDIJ09-518a-like plasmids (*Supplementary file 1*). To date, among the 53 fully sequenced *qnrD*-plasmids, ~81% are reported to be pDIJ09-518a-like plasmids and ~19% to be p2007057-like plasmids (*Figure 1*).

The SOS stress response is a key mechanism by which bacteria respond to DNA damage (*Baharoglu and Mazel, 2014*; *Erill et al., 2007*). Oxidative stress and sub-minimum inhibitory concentrations (MICs) of certain antibiotics can cause DNA damage, triggering the SOS response (*Baharoglu et al., 2013*; *Baharoglu and Mazel, 2011*; *Kohanski et al., 2010*). Once triggered, the SOS response favours bacterial survival in numerous ecological settings, including the likely emergence of antibiotic-resistant isolates in patients, when sub-MIC antibiotic concentrations occur at the infection site, through the transient increase of mutation rate accompanying the SOS response (*Matic, 2019*). Fluoroquinolones are known to induce the SOS response in *E. coli* (*Baharoglu and Mazel, 2014*; *Recacha et al., 2017*). In contrast, aminoglycosides are able to induce the SOS response in bacteria such as *Vibrio cholerae* but not in *E. coli*, which is better equipped to tackle oxidative stress (*Baharoglu et al., 2014*; *Baharoglu et al., 2013*; *Baharoglu and Mazel, 2011*). Two main mechanisms have been reported to explain why aminoglycoside-mediated oxidative stress in *E. coli* does not lead to SOS induction, and include: (1) the production of the general stress response sigma factor RpoS, which induces DNA polymerase IV, and (2) increased activity of the GO-repair system, which removes the mutagenic oxidized guanine (GO lesion) (*Michaels and Miller, 1992*), along with the base excision repair pathway (*Baharoglu et al., 2013*). Nitric oxide (NO), which can be converted to other reactive nitrogen species, can also cause DNA damage, inducing also the SOS response (*Lobysheva et al., 1999*; *Nakano et al., 2005a*). But, *E. coli* possesses the well-characterized Hmp flavohaemoprotein that detoxifies NO either by producing nitrate ($NO_3^-$) under aerobic conditions, or by a $O_2$ autoreduction to nitrous oxide ($N_2O$) (*Cruz-Ramos et al., 2002*; *Stevanin et al., 2007*).

In this study, we show that in *E. coli* carrying pDIJ09-518a-like plasmids the SOS response can nonetheless be triggered by sub-MIC levels of aminoglycosides through NO accumulation. The latter is all at once due to higher NO formation and to the repression of the Hmp-mediated detoxification pathway driven by proteins encoded by this small *qnrD*-plasmid.

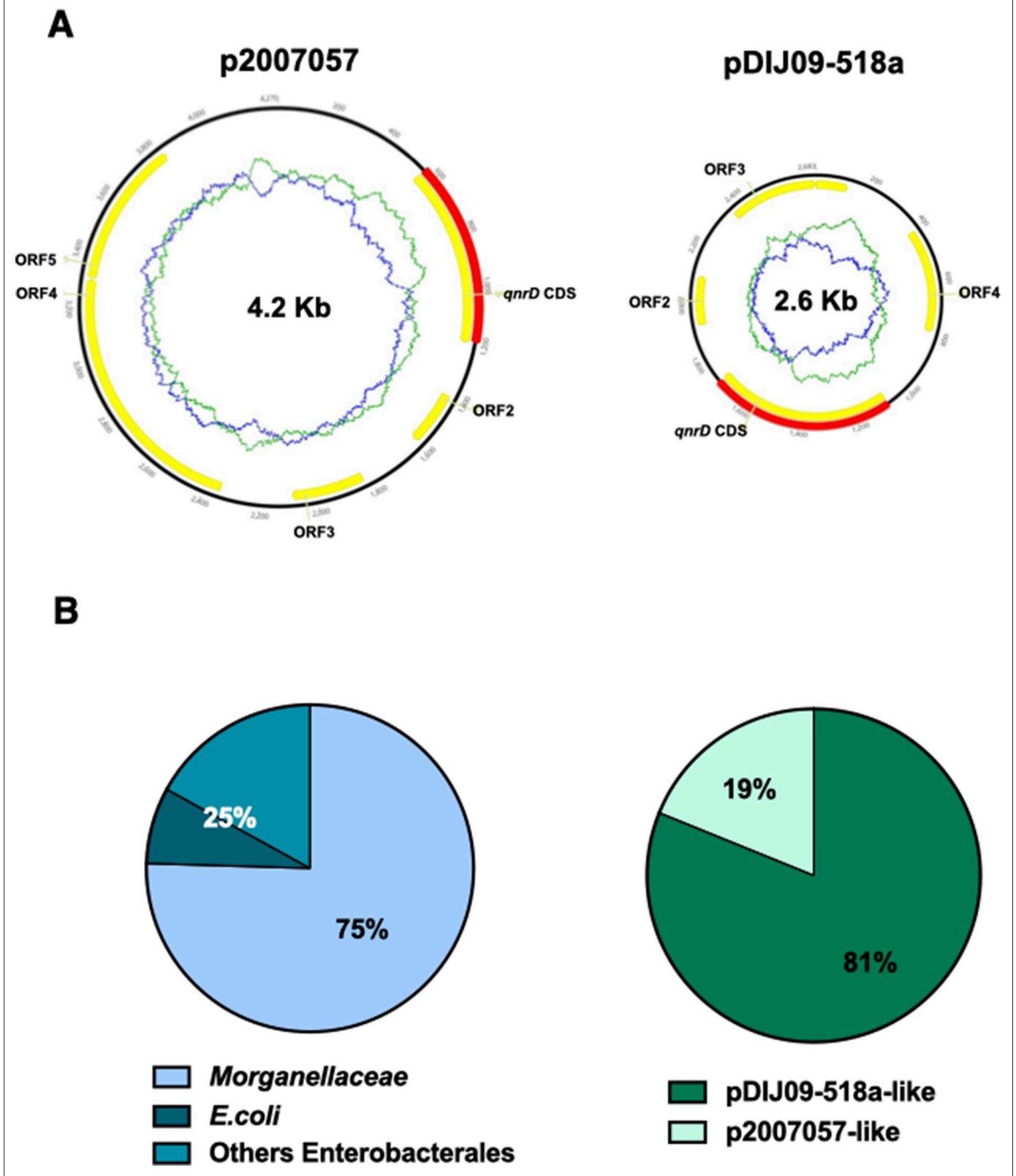

**Figure 1.** *qnrD* genes are carried by small plasmids. (**A**) The two *qnrD*-plasmid archetypes: p2007057 and pDIJ09-518a. (**B**) Distribution of the *qnrD*-plasmids among the 53 *qnrD* fully sequenced plasmids available in GenBank.

The online version of this article includes the following source data for figure 1:

**Source data 1.** qnrD genes in small plasmids.

## Results

### *qnrD* expression is SOS regulated and triggered by aminoglycosides

Transcription of the *qnrB and qnrD* genes has been shown to be controlled by the LexA-mediated SOS response. LexA represses the SOS regulon genes by binding to its cognate LexA-box or SOS-box sequence on the promoter. Thus, the LexA proteolysis leads to de-repression of this regulon, comprised of about 40 genes in *E. coli* (*Courcelle et al., 2001*).

Unlike *qnrB*, the regulation of the most recently characterized PMQR gene, *qnrD,* by the SOS response has only been partially described, given that LexA dependence has not yet been evidenced (*Briales et al., 2012*; *Da Re et al., 2009*). In *E. coli*, nucleofilament of the SOS activator RecA induce the LexA repressor self-cleavage, therefore inducing the SOS global regulatory network (*Baharoglu and Mazel, 2014*). All the genes belonging to the SOS regulon carry a SOS-box in their promoter. The SOS-box, alternatively named LexA-box, is a 16-bp long sequence recognized by the LexA repressor. To fill the knowledge gaps regarding *qnrD* regulation, we conducted a multiple alignment analysis of all the 53 *qnrD*-plasmids, fully sequenced and available in GenBank (*Supplementary file 1*).

Among these plasmids, we found a highly conserved putative SOS-box upstream of the *qnrD* start codon (see the highlighted logo consensus sequence in *Figure 2A*). This sequence is similar (15/16 identical bases) to the one found in *E. coli*.

We demonstrated the functionality of this *qnrD* SOS-box in the pDIJ09-518a plasmid and its dependence on the RecA and LexA proteins, which are both essential for the SOS response. To do this, we quantified *qnrD* gene expression levels from the native *qnrD*-plasmid, pDIJ09-518a, in the presence of sub-MICs levels of mitomycin C and ciprofloxacin, two well-known SOS inducers (see *Supplementary files 2–4*). It has been previously established that aminoglycosides do not induce the SOS response in *E. coli* (*Baharoglu et al., 2014*; *Baharoglu et al., 2013Baharoglu et al., 2013*; *Baharoglu and Mazel, 2011*) and therefore the aminoglycoside tobramycin was used as a negative control in our qRT-PCR assays. Using *E. coli* MG1656/pDIJ09-518a grown in lysogeny broth (LB, MG1656 henceforth referred to as *E. coli* in the text and WT in the figures), we found a 2.18- and 2.02-fold increase in *qnrD* expression induced by mitomycin C and ciprofloxacin, respectively. Unexpectedly, upon tobramycin treatment, we found a 2.8-fold increase in *qnrD* expression (*Figure 2B*). To confirm the role of the SOS in induced *qnrD* transcription in the presence of these three drugs, *qnrD* RNA levels were assessed in isogenic *E. coli / pDIJ09-518a* derivatives where the SOS response was not effective due to (1) deletion of its activator (Δ*recA*), (2) a mutation leading to a non-cleavable repressor (*lexAind*), or (3) inactivation of the *qnrD* SOS-box directly on pDIJ09-518a (LexA-box*: wild-type sequence: CTGTATA-AATAACCAG; modified SOS-box: AGCTATAAATAACCAG) (*Figure 2B*). As expected, in strains where the SOS response was blocked, *qnrD* expression was not increased by ciprofloxacin or mitomycin C treatments, but this response was also blocked in the presence of tobramycin. To conclusively assert that RecA is needed to increase *qnrD* expression by SOS-dependent regulation, we showed that *qnrD* expression upon tobramycin exposure was increased in the *recA* mutant complemented strain, comparable to the levels detected in the *E. coli*/pDIJ09-518a strain (*Figure 2B*).

As shown in *Figure 2—figure supplement 1A*, by determining the growth of both *E. coli* and *E. coli* carrying the *qnrD*-plasmid, we showed that sub-MIC tobramycin treatment has no impact on the viability of *E. coli*/pDIJ09-518a. To assess the stability of pDIJ09-518a carriage in the absence of a fluoroquinolone selective pressure, we performed daily iterative subcultures onto fresh LB media for 30 days (*Figure 2—figure supplement 1B, C*), using agar plates with or without 0.06 µg/ml ciprofloxacin. We quantified bacterial viability over this period and confirmed the maintenance of the *qnrD*-plasmid by PCR, every 5 days (five individual colonies) and found no differences in these parameters over the 30-day period of observation.

To further explore our finding that the SOS response could be triggered by aminoglycosides in *E. coli* carrying pDIJ09-518a, we quantified the expression of *sulA,* a well-known SOS regulon-induced gene (*Huisman and D'Ari, 1981*), in the presence of aminoglycosides. As shown in *Figure 2C*, neither tobramycin nor gentamicin were able to increase *sulA* expression in *E. coli*. In cells harbouring pDIJ09-518a, however, a 3.13- and 3.71-fold change in *sulA* expression was measured in the presence of sub-MICs of tobramycin and gentamicin, respectively, confirming that aminoglycosides can induce the SOS response in *qnrD*-plasmid-bearing *E. coli* strains. We also confirmed that pDIJ09-518a did not induce the SOS response in the absence of aminoglycosides in *E. coli* (*Figure 2D*). As the pDIJ09-518a-like plasmids are found mostly in *Morganellaceae*, the *sulA* gene expression was quantified

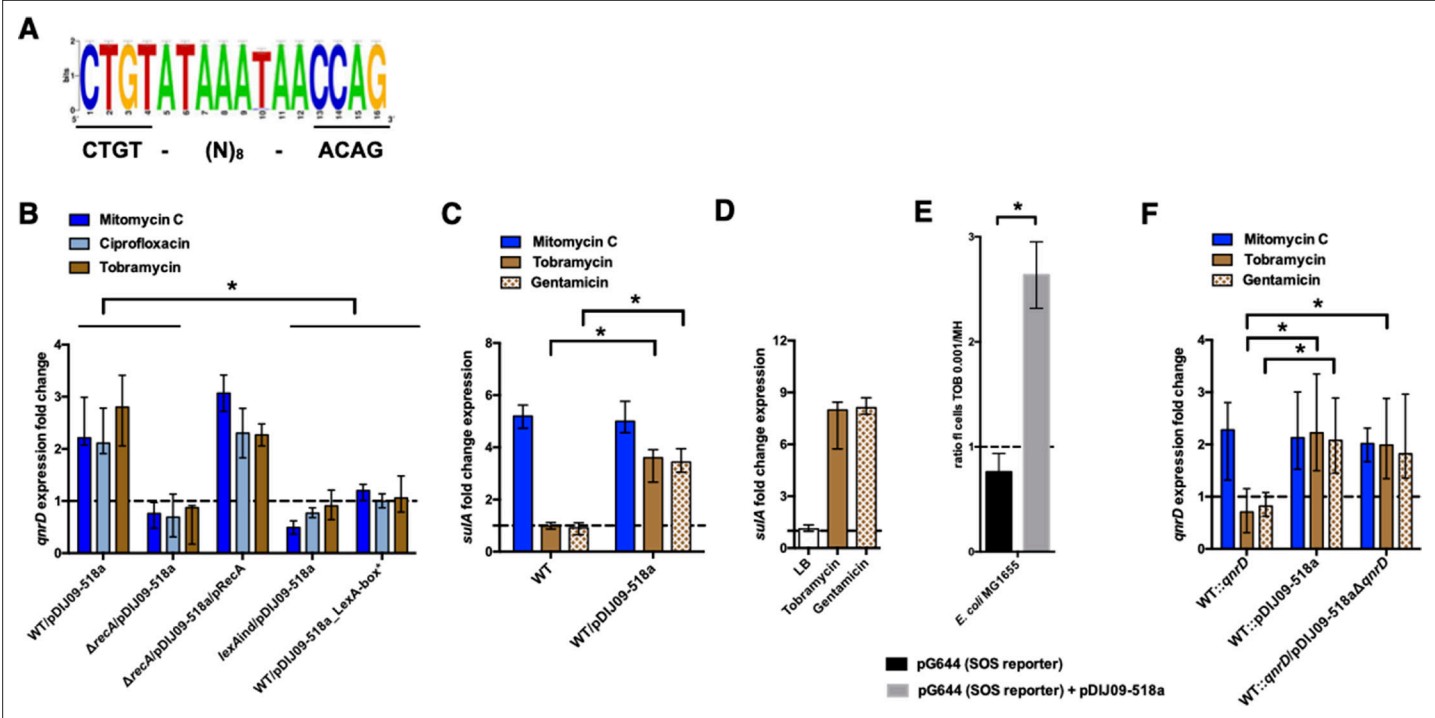

**Figure 2.** *qnrD* regulation is SOS-mediated and aminoglycosides induce the SOS in *E. coli* because of the *qnrD*-plasmid backbone. (**A**) The *qnrD* SOS-box conservation by visualization of the consensus sequence logo generated from the 53 fully *qnrD*-plasmid sequences. The consensus sequence for *E. coli* is indicated below. (**B**) Relative expression of *qnrD* in *E. coli* MG1656 (WT) derived isogenic strains carrying pDIJ09-518a or pDIJ09-518a with a modified *qnrD*-SOS-box (LexA-box*), exposed to mitomycin C (dark blue), ciprofloxacin (blue), or tobramycin (brown) in comparison to expression in lysogeny broth (LB), normalized with *dxs*. (**C**) Relative expression of *sulA* in *E. coli* MG1656 (WT) with or without pDIJ09-518a, exposed to mitomycin C (dark blue), tobramycin (brown), or gentamicin (dotted brown) in comparison to expression in LB, normalized with *dxs*. (**D**) Relative expression of *sulA* in *E. coli* MG1656 (WT) with pDIJ09-518a, grown in LB, or either with tobramycin or with gentamicin, in comparison to the expression in *E. coli* MG1656, in the three culture conditions, and normalized with *dxs*. (**E**) Histogram bars show the ratio of green fluorescent protein (GFP) fluorescence in a *E. coli* MG1655 WT carrying or not the pDIJ09-518a plasmid in the presence of tobramycin (0.001 µg/ml) over fluorescence of the same strain grown in Mueller-Hinton (MH) reflecting induction of SOS. Black bars stand for strain with the SOS reporter vector and grey bars stand for strain carrying the *qnrD*-plasmid pDIJ09-518a and the SOS reporter vector. (**F**) Relative expression of *qnrD* in *E. coli* MG1656 with *qnrD* and its own promoter, or the native *qnrD*-plasmid inserted into the chromosome, or chromosomal *qnrD* complemented with pDIJ09-518aΔ*qnrD*, exposed to mitomycin C (dark blue), tobramycin (brown), or gentamicin (dotted brown) in comparison to expression in LB, normalized with *dxs*. Data represent median values of six independent biological replicates, and error bars indicate upper/lower values. *p < 0.05. Wilcoxon matched-pairs signed-rank test.

The online version of this article includes the following source data and figure supplement(s) for figure 2:

**Source data 1.** Relative expression of qnrD in *E. coli* MG1656 (WT) and isogenic strains.

**Source data 2.** Relative expression of sulA in *E. coli* MG1656 (WT) and isogenic strains.

**Source data 3.** Relative expression of sulA in *E. coli*/pDIJ09-518a.

**Source data 4.** GFP fluorescence in a *E. coli* MG1655 WT.

**Source data 5.** Relative expression of qnrD in *E. coli* MG1656 and isogenic strains with chormosomal complementation.

**Figure supplement 1.** The viability of *E. coli*.

**Figure supplement 1—source data 1.** OD600 measured for *E. coli* MG1656 (WT) and isogenic strains.

**Figure supplement 2.** *qnrD*-plasmid carriage does not promote the SOS response induction by tobramycin in *Providencia rettgeri*.

**Figure supplement 2—source data 1.** Relative expression of sulA in P. rettgeri/pDIJ09-518a.

in the *Providencia rettgeri* harbouring the pDIJ09-518a. As shown in *Figure 2—figure supplement 2A*, tobramycin was not able to induce *sulA* expression, whereas the SOS induction was observed in cells exposed to mitomycin C. It seems that the burden presented by the *qnrD*-plasmid in *E. coli* is not present in *Providencia* spp. We conducted a protein sequence alignment of D4C4 × 8_PRORE (UniProt annotation of *P. rettgeri* Hmp *protein*) using BLASTp with the protein Hmp from *E. coli*. The results of alignments showed a 63,38% protein identity with *E. coli* (*Figure 2—figure supplement 2B*).

The lack of SOS induction may result of Hmp from *Providencia* that could be not inhibited by ORF4 due to this incomplete identity, allowing then Hmp to play its role in NO detoxification. However, we cannot rule out another hypothesis with ORF3 inactive in *Providencia* spp. leading to less NO formation that in *E. coli*.

To corroborate SOS-mediated *qnrD* expression with exposure to aminoglycosides, we looked at SOS induction in *E. coli* (MG1655 for these experiments) using a previously published SOS reporter setup (*Baharoglu et al., 2010*). In this system, a GFP-encoding gene is put under the control of the well-characterized SOS-driven *recN* promoter (plasmid pG644, *Supplementary file 3*; *Baharoglu et al., 2010*), and GFP fluorescence, measured by flow cytometry, gives a readout of the SOS induction level. Using this, we confirmed that, in the presence of aminoglycosides, the SOS response was induced in *E. coli* MG1655 carrying pDIJ09-518a (2.6-fold higher), while no increase in fluorescence was observed in WT *E. coli* MG1655 (*Figure 2E*).

## The plasmid backbone contributes to the increase of the *qnrD* gene expression upon aminoglycosides exposure

To determine which components of the pDIJ09-518a plasmid contributed to the SOS response induction in the presence of sub-MIC aminoglycoside treatments, we inserted into the chromosome of *E. coli*, in the *cynX* and *lacA* chromosome intergenic region, either the *qnrD* gene with its own promoter (WT::*qnrD*) alone or the entire plasmid (WT::pDIJ09-518a) (see *Supplementary files 3 and 4*). This intergenic region between *cynX* and *lacA* tolerates the insertion by homologous recombination in vitro and in vivo, and thus this locus is neutral for the fitness of the bacteria (*Warr et al., 2019*).

For both strains, *qnrD* expression was increased (2.27- and 2.13-fold) in response to treatment with mitomycin C, confirming that neither *qnrD* nor pDIJ09-518a insertion into the *E. coli* chromosome had a negative effect on the SOS induction pathway (*Figure 2F*). However, in the presence of sub-MICs levels of tobramycin and gentamicin, no change in *qnrD* transcripts was found in *E. coli* WT::*qnrD*, whereas when the entire plasmid was inserted into the chromosome, *qnrD* transcript levels were increased upon both tobramycin and gentamicin exposure (2.22- and 2.08-fold increase, respectively) (*Figure 2F*). Finally, in a WT::*qnrD* strain complemented with the *qnrD*-deleted-pDIJ09-518a plasmid (pDIJ09-518aΔ*qnrD*), exposure to sub-MICs levels of tobramycin- or gentamicin-induced *qnrD* transcription from the chromosomal site to levels similar to those observed for the WT::pDIJ09-518a strain. Altogether, these results confirmed that in a *qnrD*-harbouring *E. coli* strain, the pDIJ09-518a plasmid backbone without the *qnrD* gene is sufficient to elicit an SOS response by aminoglycosides.

## Small *qnrD*-plasmid promotes nitrosative stress in *E. coli*

In *E. coli*, reactive oxygen species (ROS) are well-known inducers of the SOS response, but as mentioned above, exposure to sub-MIC of aminoglycosides does not lead to ROS accumulation because of the high-level stability of the RpoS (*Baharoglu et al., 2013*). We hypothesized that carrying *qnrD*-plasmid pDIJ09-518a could affect this stability and therefore increase ROS, causing SOS induction. Thus, we tested the effect of tobramycin on oxidative stress in *E. coli* or *E. coli*/pDIJ09-518a (*Figure 3—figure supplement 1*). Dihydrorhodamine 123 (DHR) oxidation detects the presence of hydrogen peroxide ($H_2O_2$), resulting in the dismutation of the superoxide anion, which is reduced into hydroxyl radicals according to the Fenton reaction (*Henderson and Chappell, 1993*). Then, fluorescence is measured as an indicator of ROS generation. For this assay, ciprofloxacin was used as a positive inducer of ROS formation. We found no ROS formation induced by tobramycin in either *E. coli* or *E. coli*/pDIJ09-518a (*Figure 3A*, ratio ~1, brown bar). As previously reported by Machuca et al. for other PMQR determinants (*Machuca et al., 2014*), we did not find increased ROS formation in *E. coli* carrying the *qnrD*-plasmid upon exposure to sub-MIC of ciprofloxacin (*Figure 3A*, ratio ~1, blue bar). However, the underlying mechanism explaining these findings has yet to be identified.

The catalase KatG is a key player in *E. coli* by scavenging $H_2O_2$ and thereby limits accumulation $H_2O_2$ (*Triggs-Raine et al., 1988*). We used *katG* expression as an $H_2O_2$ transcriptional reporter (*Supplementary file 4*) in order to confirm results from the ROS-detecting dye-based assay.

We found increased expression of *katG*-induced by $H_2O_2$ in both the WT strain or plasmid-bearing *E. coli*/pDIJ09-518a (*Figure 3B*, ratio ~3 and 2, white bars). But interestingly, we did not find any increased *katG* expression induced by tobramycin in either *E. coli* or *E. coli*/pDIJ09-518a (*Figure 3B*,

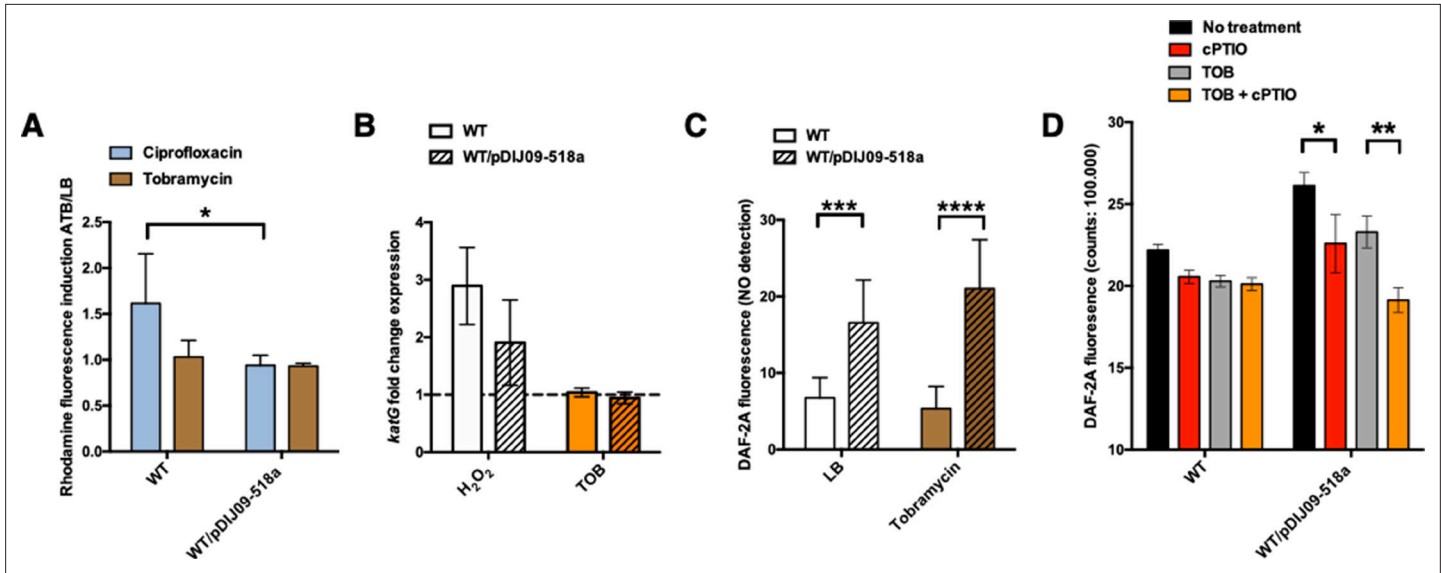

**Figure 3.** Small *qnrD*-plasmid promotes nitrosative stress in *E.coli*. (**A**) Reactive oxygen species (ROS) formation for *E. coli* MG1656 (WT) and its derivative carrying pDIJ09-518a cultured in lysogeny broth (LB) or exposed to tobramycin. Production of ROS was calculated as the mean ratio of the dihydrorhodamine 123 (DHR-123) fluorescence of the treated samples to the control samples (*n* = 6). Data were analysed using a two-way analysis of variance (ANOVA) with an p value <0.05 for strains as a source of variation in the overall ANOVA. *p < 0,05 using Tukey's multiple comparisons test. Mean difference for WT compared to WT/pDIJ09-518a exposed to ciprofloxacin was 0.6753; 95% confidence interval (CI) of difference [0.07565; 1.294]. Error bars represent the standard deviation (SD). (**B**) Relative expression of *katG* in *E. coli* MG1656 (WT) and in *E. coli* MG1656 carrying pDJJ09-518a, in comparison to expression in LB, normalized with *dxs*. Data represent median values of six independent biological replicates and error bars indicate upper/lower values. Wilcoxon matched-pairs signed-rank test. (**C**) Nitric oxide (NO) formation for *E. coli* MG1656 (WT) and its derivative carrying pDIJ09-518a culture in LB or exposed to tobramycin. Production of NO was calculated as the mean ratio of the DAF-2A fluorescence (*n* = 9). *y*-Axis represents arbitrary units of fluorescence. Data were analysed using a two-way ANOVA with a p value <0.0001 for strains as a source of variation in the overall ANOVA. ***p < 0.001 and ****p < 0.0001 using a Tukey's multiple comparisons test. The mean difference for WT compared to WT/pDIJ09-518a grown in LB was −9.790 [95% CI, −15.75; −3.825]. The mean difference for WT compared to WT/pDIJ09-518a exposed to tobramycin was −15.66 [95% CI, −21.62; −9.695]. Error bars represent the SD. Data represent median values of four independent biological replicates, and error bars indicate upper/lower values. Wilcoxon matched-pairs signed-rank test. (**D**) Histogram bars show the DAF-2A fluorescence, in *E. coli* WT and WT/pDIJ09-518a as a measure of intracellular in NO obtained using a FACS-based approach, with or without NO scavenger (carboxy-PTIO, cPTIO). Data were analysed using a two-way ANOVA with a p value <0.0001 for treatment as a source of variation in the overall ANOVA. *p < 0.05 and **p < 0.01 using a Tukey's multiple comparisons test. The mean difference for WT/pDIJ09-518a grown in LB compared to WT/pDIJ09-518a grown in LB and cPTIO was 3.545 [95% CI, 0.4866; 6.603]. The mean difference for WT/pDIJ09-518a grown with tobramycin (TOB) compared to WT/pDIJ09-518a grown with TOB and cPTIO was 4.160 [95% CI, 1.163; 7.157]. Error bars represent the SD. Data represent median values of three independent biological replicates, and error bars indicate upper/lower values.

The online version of this article includes the following source data and figure supplement(s) for figure 3:

**Source data 1.** DHR-123 fluorescence for ROS formation in *E. coli* MG1656 (WT) and isogenic strains.

**Source data 2.** Relative expression of katG in *E. coli* MG1656 (WT) and isogenic strains.

**Source data 3.** DAF-2A fluorescence for NOS formation in *E. coli* MG1656 (WT) and isogenic strains.

**Source data 4.** DAF-2A fluorescence obtained using a FACS-based approach, in *E. coli* WT and isogenic strains.

**Figure supplement 1.** Schematic approach for fluorometric detection of free intercellular reactive oxygen species (ROS) and nitric oxide (NO).

ratio ~1, orange bars), confirming that SOS induction in *E. coli* carrying the *qnrD*-plasmid exposed to tobramycin is not due to increased intracellular ROS.

In an effort to find another SOS inducer, we measured the intracellular production of NO by determining 5,6-diaminofluorescein diacetate (DAF-2 DA) fluorescence (***Figure 3—figure supplement 1***; ***Kojima et al., 1998***; ***Lobysheva et al., 1999***). Strikingly, NO production was significantly elevated in the strains carrying the *qnrD*-plasmid in the absence of an aminoglycoside stimulus, and only increased slightly further when the antibiotic was added at sub-MIC levels (16.55 when strains were grown in LB and an increase of up to 21.01 in the presence of tobramycin) (***Figure 3C***). To confirm that the NO production quantified in *E. coli*/pDIJ09-518a was due to the plasmid carriage, we quantified NO production with the addition of a NO scavenger (carboxy-PTIO, cPTIO) (***Figure 3D***). NO levels were

decreased in the presence of cPTIO. These results strongly suggest that the carriage of the *qnrD*-plasmid by itself induces a nitrosative stress in *E. coli*.

## The GO-repair system is involved in the aminoglycoside-induced SOS response in *E. coli* carrying small *qnrD*-plasmid

Considering that the *qnrD*-plasmid-mediated NO and the aminoglycosides seem to be prerequisites to the SOS response induction in *E. coli*/pDIJ09-518a, we considered that the SOS response could results from the deleterious effects common to both these stimuli. Aminoglycosides, as well as NO and other reactive nitrogen species, such as dinitrogen trioxide ($N_2O_3$) and peroxynitrite ($ONOO^-$), form 8-oxo-G (7,8-dihydro-8-oxoguanine) (*Baharoglu et al., 2013*; *Nakano et al., 2005a*; *Nakano et al., 2005b*). It has been described that the incorporation of 8-oxo-G into DNA leads to A/G mismatch during replication (*Grollman and Moriya, 1993*; *Michaels et al., 1992*; *Michaels and Miller, 1992*). To prevent or repair these types of oxidative lesions, *E. coli* uses the GO-repair system, with MutT and MutY as two of the key players in this error-prevention system (*Baharoglu et al., 2014*; *Boiteux et al., 2017*; *David et al., 2007*; *Foti et al., 2012*; *Grollman and Moriya, 1993*; *Maki and Sekiguchi, 1992*; *Michaels et al., 1992*). Incomplete action of the GO-repair system leads to the formation of double strand DNA breaks (DSBs), which are lethal if unrepaired (*Foti et al., 2012*). MutT acts as a nucleotide

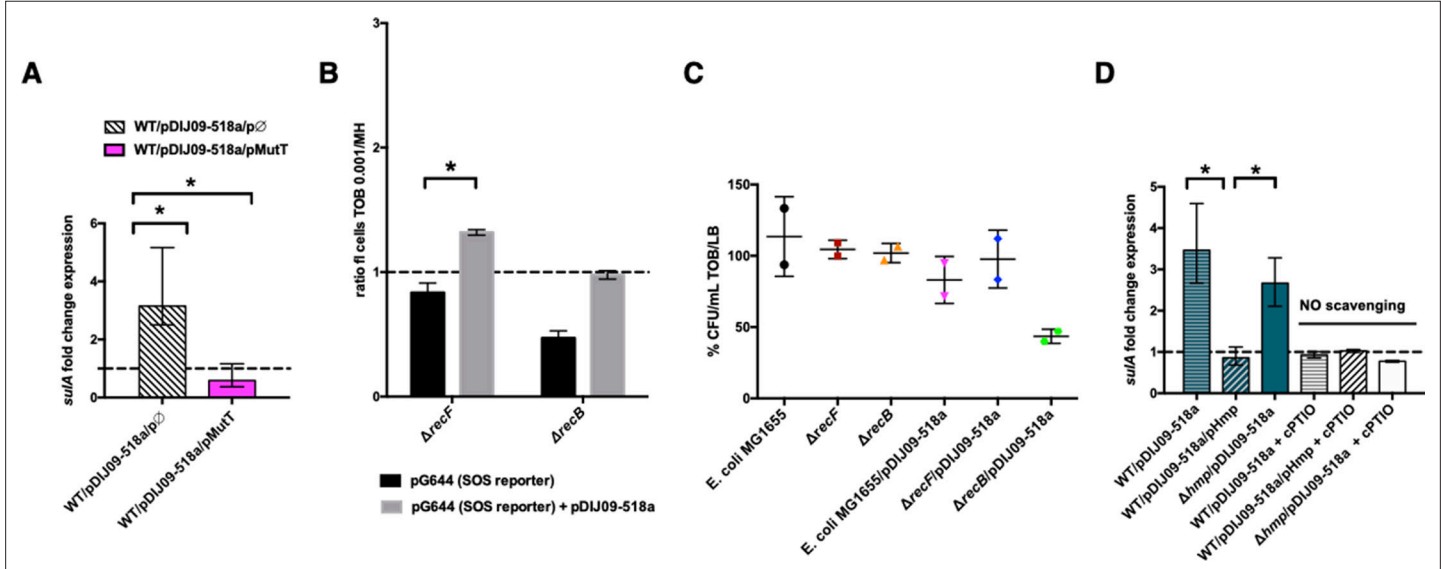

**Figure 4.** Aminoglycosides induce SOS in *E. coli*/pDIJ09-518a due to overwhelmed GO-repair pathway associated with inactivated Hmp. (**A, D**) Relative expression of *sulA* in *E. coli* MG1656 (WT) isogenic strains carrying pDJJ09-518a, overexpressing the GO-repair system protein MutT and the *hmp*-deleted mutant, exposed to tobramycin, treated with the nitric oxide (NO) scavenger carboxy-PTIO (cPTIO) (for D), in comparison to expression in lysogeny broth (LB), normalized with *dxs*. Data represent median values of six independent biological replicates and error bars indicate upper/lower values. Wilcoxon matched-pairs signed-rank test. (**B**) Histogram bars show the ratio of GFP fluorescence in a *E. coli* MG1655 Δ*recB* and Δ*recF* in the presence of tobramycin (0.001 μg/ml) over fluorescence of the same strain grown in MH reflecting induction of SOS. Black bars stand for strain with the SOS reporter vector and grey bars stand for strain carrying the *qnrD*-plasmid pDIJ09-518a and the SOS reporter vector. (**C**) Impact of *recB* gene inactivation in *E. coli* harbouring the *qnrD*-plasmid on growth in sub-MIC tobramycin. Histogram bars represent the percentage of the ratio of colony-forming units (CFUs)/ml for each strain in tobramycin (0.001 μg/ml) over CFU/ml in LB. Data represent median values of two independent biological replicates, and error bars indicates the standard deviation (SD). Wilcoxon matched-pairs signed-rank test. *p < 0.05.

The online version of this article includes the following source data and figure supplement(s) for figure 4:

**Source data 1.** Relative expression of sulA in *E. coli* MG1656/ pDJJ09-518a overexpressing MutT.

**Source data 2.** GFP fluorescence in a *E. coli* MG1655 ΔrecB and ΔrecF.

**Source data 3.** Ratio of colony-forming units.

**Source data 4.** Relative expression of sulA in *E. coli* MG1656/pDJJ09-518a with deleted hmp.

**Figure supplement 1.** *hmp* deletion and empty vector carriage do not promote the SOS response induction.

**Figure supplement 1—source data 1.** Relative expression of sulA in *E. coli* MG1656 Δhmp and *E. coli* MG1656 carrying empty vector.

**Figure supplement 1—source data 2.** Relative expression of sulA in hmp mutant and derivatives strains.

pool sanitizer removing 8-oxo-G triphosphate, while MutY removes the adenine base from 8-oxo-G/A mispairing. We further hypothesized that the SOS response could rely on the deleterious effect of unremoved oxidized guanines due to the combined stress of NO and aminoglycosides.

To test the hypothesis that the SOS response induced by aminoglycosides is linked to the formations of 8-oxo-G-mediated DSBs because of their incomplete removal, we measured the SOS response in *E. coli* carrying pDIJ09-518a with or without overexpression of MutT (*Supplementary files 3 and 4*). As shown in *Figure 4A*, we found that the SOS response, measured through *sulA* transcription levels, was induced in the presence of aminoglycosides in *E. coli*/pDIJ09-519a carrying the empty vector used for MutT overexpression (3.15-fold increase), while it was not increased in *E. coli*/pDIJ09-518a overexpressing MutT (0.59-fold change).

To further determine if the incomplete action of the GO-repair system may lead to the accumulation of DSBs in *E. coli*/pDIJ09-518a exposed to aminoglycosides, we used two *E. coli* MG1655 strains wherein the SOS could either be induced or not activated in the presence of DSBs. The RecFOR pathway allows RecA nucleo-filament formation on single-stranded DNA breaks, whereas the RecBCD recruits RecA onto DSBs (*Baharoglu and Mazel, 2014*; *Kuzminov, 1999*). As shown in *Figure 4B*, SOS was not induced by aminoglycosides in the *recB*-deleted strain, whereas a slight induction of the SOS response was detected in the *recF* deletant (1.4-fold change compared to growth in LB). These results suggest that in *E. coli* carrying *qnrD*-plasmids, DSBs are produced upon aminoglycoside treatment inducing the SOS response necessary for bacterial survival. The deletion of *recB* in *E. coli* harbouring the plasmid causes loss of viability after exposure to tobramycin as compared to the wild-type *E. coli* MG1655 or the *recF* mutant (*Figure 4C*).

## Role of the NO-detoxifying Hmp in the SOS induction upon exposure to sub-MIC of aminoglycosides

In *E. coli*, the Hmp flavohaemoprotein has been described as the key player in detoxifying NO under aerobic conditions (*Poole et al., 1996*). We therefore hypothesized that NO accumulation could be due to ineffective Hmp-mediated detoxification leading to the accumulation of 8-oxo-G and 8-nitro-G DNA lesions and thereby, an SOS response induction in *E. coli* exposed to aminoglycosides. To test this hypothesis, we overexpressed Hmp in *E. coli*/pDIJ09-518a (*Supplementary files 3 and 4*). As shown in *Figure 4D*, the *sulA* transcription increase observed in *E. coli*/pDIJ09-518a in the presence of aminoglycosides (3.14-fold increase over growth without aminoglycosides) was now abolished when Hmp was overexpressed (0.85-fold decrease compared to growth without aminoglycosides). To confirm this finding, we quantified the SOS induction in a *qnrD*-plasmid-harbouring *E. coli* strain where *hmp* had been deleted. In this strain, unable to detoxify NO, the carriage of the *qnrD*-plasmid increased the expression of *sulA* (~2.5-fold) when exposed to tobramycin (*Figure 4D*). We confirmed that neither deleting *hmp* nor carrying the empty vector used to overexpress Hmp, increased the expression of *sulA* in *E. coli* grown in LB medium without aminoglycosides (*Figure 4—figure supplement 1A*). Furthermore, the addition of tobramycin did not induce the SOS response in the *hmp* mutant (*Figure 4—figure supplement 1B*). After exposure to tobramycin, complementation of the *hmp* mutant carrying the *qnrD*-plasmid with pHmp (a plasmid expressing *hmp*), did not trigger the SOS response, whereas the SOS was induced in the same genetic context but no *hmp* complementation (*Figure 4—figure supplement 1B*). To confirm our hypothesis, we quantified SOS induction in the same derivative strains exposed to aminoglycosides using a NO scavenger (cPTIO) assay. It is noteworthy that in the case without any NO, the SOS response was not induced (*Figure 4D*).

Our results show that *qnrD*-plasmid-bearing *E. coli* undergoes nitrosative stress because of a much less effective Hmp-mediated detoxification that leads to SOS response induction when exposed to aminoglycosides. These findings underscore that both NO accumulation and tobramycin are needed for the induction of the SOS response in *E. coli* carrying pDIJ09-518a.

## Small *qnrD*-plasmid genes promote NO formation and inhibit NO detoxification

Next, we tried to establish which pDIJ09-518a ORF(s) promoted the NO accumulation leading to the tobramycin-induced SOS response in *E. coli*. Among bacteria, NO synthesis by NO synthase has been seldomly reported, that is in *Nocardia* spp, though this bacterial enzyme is different from the mammalian version. In most other bacteria, however, the main sources of NO come from the activity

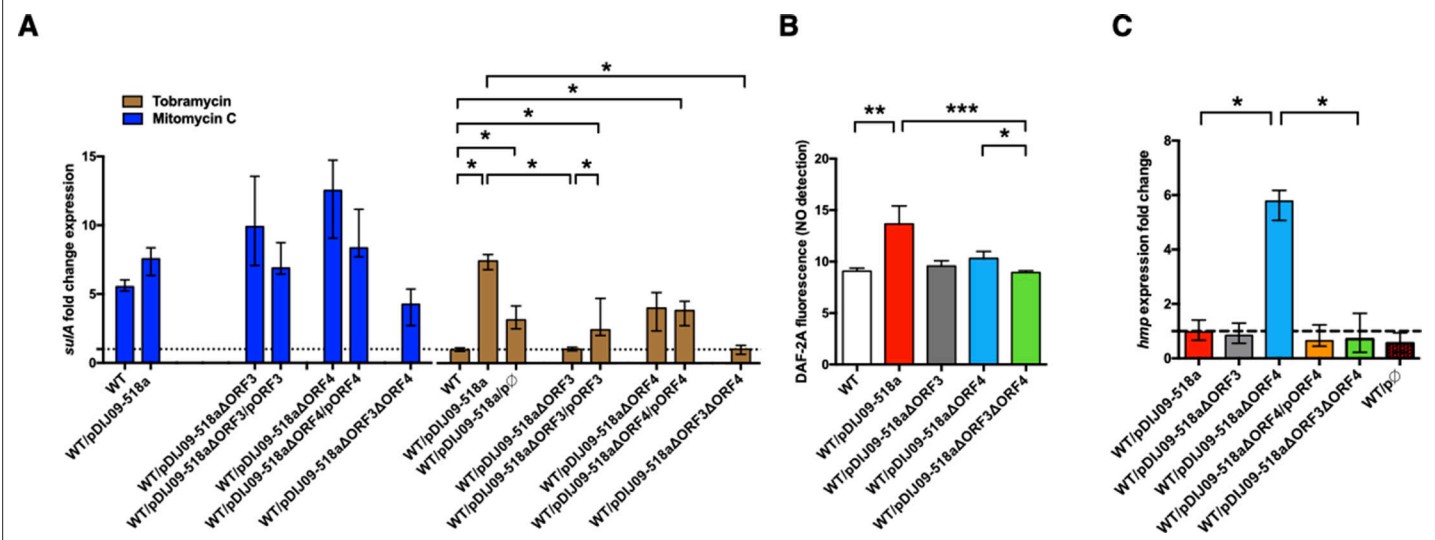

**Figure 5.** Deletion of ORF3 decreases the SOS response induction, after tobramycin treatment and ORF4 regulates the Hmp nitric oxide detoxification pathway. (**A**) Relative expression of *sulA* in *E. coli* MG1656 (WT) derived isogenic strains carrying pDIJ09-518a with ORF3 and/or ORF4 deleted and complemented, exposed to mitomycin C (dark blue) or tobramycin (brown) in comparison to expression in lysogeny broth (LB), normalized with *dxs*. Data represent median values of six independent biological replicates, and error bars indicate upper/lower values. *p < 0.05. Wilcoxon matched-pairs signed-rank test. (**B**) Nitric oxide (NO) formation for the isogenic strains (*n* = 6). Data were analysed using a Kruskal–Wallis test, with a p value <0.0001 for the overall analysis of variance (ANOVA). NO formation for each strain was analysed using Dunn's multiple comparisons test. *p < 0.05, **p < 0.01, and ***p < 0.001. Bars represent mean values and SD. (**C**) Relative expression of *hmp* in *E. coli* MG1656 (WT) derivative isogenic strains carrying pDIJ09-518a with ORF3 and/or ORF4 deleted and complemented, or the empty vector, grown in LB, in comparison to expression in *E. coli* MG1656 (WT), normalized with *dxs*. Data represent median values of six independent biological replicates, and error bars indicate upper/lower values. Wilcoxon matched-pairs signed-rank test. *p < 0.05.

The online version of this article includes the following source data and figure supplement(s) for figure 5:

**Source data 1.** Relative expression of sulA in *E. coli* MG1656 (WT) derived isogenic strains #1.

**Source data 2.** Relative expression of sulA in *E. coli* MG1656 (WT) derived isogenic strains #2.

**Source data 3.** Relative expression of sulA in *E. coli* MG1656 (WT) derived isogenic strains #3.

**Source data 4.** DAF-2A fluorescence in *E. coli* MG1656 (WT) and complemented strains.

**Source data 5.** Relative expression of hmp in *E. coli* MG1656 (WT) and isogenic strains.

**Figure supplement 1.** Spontaneous mutation ratio after treatment with sub-minimum inhibitory concentration (MIC) of tobramycin.

**Figure supplement 1—source data 1.** CFU counting after treatment of sub-MIC of tobramycin.

of nitrite and nitrate reductases, which catalyse the reduction of nitrate ($NO_3^-$) and/or nitrite ($NO_2^-$) to NO (*Crane et al., 2010*). Querying the pBLAST/psiBLAST databases, we found that ORF3 encodes a putative flavin adenine dinucleotide (*FAD*)-binding oxidoreductase with a NAD(P)-binding Rossmann-fold (27% protein identity over 57% of the FAD-binding oxidoreductase domain from an *Erythrobacter* sp. H100200 accession name WP_067505159.1 and NCBI HMM evidence accession ID NF013714.2, *E* value 2e−03) that could be involved in NO production. We further found that the adjacent ORF4 encodes a putative cAMP-receptor *protein* (*CRP*)/fumarate and nitrate reduction regulatory protein (*FNR*)-type transcription factor (CRP/FNR; 24% protein identity over 71% of CRP/FNR from *Pedobacter panaciterrae* accession WP068888645.1 with the evidence accession ID 11429533 in NCBI SPARCLE, *E* value 5e−04), an $O_2$-responsive regulator of *hmp* expression.

We hypothesized that ORF3 could be involved in the aminoglycoside-induced SOS response by promoting NO production, which was tested by the deletion of ORF3 from pDIJ09-518a (*E. coli*/pDIJ09-518aΔORF3) (*Supplementary files 3 and 4*). ORF3 deletion led to the loss of *sulA* induction in response to tobramycin (*Figure 5A*, brown bar). Complementation by ORF3 (*E. coli*/pDIJ09-518aΔORF3/pORF3) restored the SOS response (*Figure 5A*). It is likely that this response is lower (but statistically significant) in this complemented strain than in *E. coli*/pDIJ09-518a given the effect on the SOS response by the empty vector pΦ used for complementations (*Figure 4—figure supplement*

*1A*). In addition, we showed that the deletion of ORF3 decreased NO production compared to that measured in the presence of the native *qnrD*-plasmid in *E. coli* (*Figure 5B*).

To determine the possible effect of the FNR-like $O_2$-response regulator encoded by ORF4 in pDIJ09-518a, on *hmp* expression (*Cruz-Ramos et al., 2002*; *Poole et al., 1996*), we deleted and complemented this ORF in pDIJ09-518a (*Supplementary files 3 and 4*). We found that ORF4 did not completely abolish the SOS response after exposure to aminoglycoside sub-MICs, as measured by *sulA* expression (*Figure 5A*). However, we did find that ORF4 deletion decreased intracellular NO production (*Figure 5B*). We next found that ORF4 deletion increased the transcription of *hmp* (*Figure 5C*) while *hmp* transcription in *E. coli*/pDIJ09-518aΔORF4 complemented with ORF4 (*E. coli*/pDIJ09-518aΔORF4/pORF4) was similar to that in *E. coli*/pDIJ09-518a. All together our results confirmed that the carriage of the expression vector alone did not impact the transcription of *hmp* compared to the parental *E. coli* in LB (*Figure 5C*).

Finally, in the presence of a double deletion in pDIJ09-518a (ΔORF3ΔORF4) (*Supplementary files 3 and 4*), we found no SOS induction by tobramycin (*Figure 5A*), a slight decrease in NO formation (*Figure 5B*) and no *hmp* expression (*Figure 5B*), confirming that only ORF4 impacts the level of *hmp* expression. Notably, neither the deletion of ORF3 nor ORF4 alleviated the SOS response induced by mitomycin C (*Figure 5A*, dark blue bars). Significant experimental effort, beyond the scope of this initial report, will be required to biochemically and genetically characterized the activities of purified proteins encoded by the ORF3 and ORF4 genes and their possible role in NO formation and detoxification under aerobic conditions.

## Aminoglycosides increased *qnrD* gene expression and promote high-level fluoroquinolone resistance

To investigate the influence of the aminoglycoside-mediated SOS induction of *qnrD* on the level of quinolone resistance, we determined the minimal inhibitory concentrations (MICs) of nalidixic acid, ciprofloxacin, ofloxacin, and levofloxacin for *E. coli* and *E. coli*/pDIJ09-518a exposed to sub-MICs of ciprofloxacin or tobramycin. Ciprofloxacin, a well-known inducer of the SOS response, was used as a control for the induction of quinolone resistance. As previously reported by *Da Re et al., 2009*, the isolates remained susceptible to quinolones but an increase in the MICs was noted. The MICs of levofloxacin, ofloxacin and ciprofloxacin were, respectively 1.3-, 1.5-, and 2-fold higher for *E. coli*/pDIJ09-518a exposed to a sub-MIC of ciprofloxacin in contrast to cells grown without antibiotic (*Table 1*).

**Table 1.** Minimal inhibitory concentration of quinolones.

| Strain* | MIC (µg/ml)† | | | | | | | | | | |
|---|---|---|---|---|---|---|---|---|---|---|---|
| | NAL | | LVX | | | OFX | | | CIP | | |
| *E. coli* MG1656 | 3 | S | 0.023 | S | | 0.006 | S | | 0.004 | S | |
| *E. coli* MG1656/pDIJ09-518a | >256 | R | 0.19 | S | | 0.25 | S | | 0.094 | S | |
| | | | | | | | | | | | |
| *E. coli* MG1656 + CIP | 2 | S | 0.023 | S | | 0.006 | S | | 0.004 | S | |
| *E. coli* MG1656/pDIJ09-518a + CIP | >256 | R | 0.25 | S | ×1.3 | 0.38 | I | x 1.5 | 0.19 | S | ×2 |
| | | | | | | | | | | | |
| *E. coli* MG1656 + TOB | 2 | S | 0.032 | S | | 0.006 | S | | 0.006 | S | |
| *E. coli* MG1656/pDIJ09-518a + TOB | >256 | R | 0.38 | S | ×2 | 0.38 | I | x 1.5 | 0.25 | S | ×2.7 |

*+CIP and +TOB stand for strains exposed to sub-MIC of ciprofloxacin and tobramycin, respectively, prior to MIC assessment.

†Susceptibility testing categories according to EUCAST clinical breakpoints. Nalidixic acid: R > 16 µg/ml. Levofloxacin: S ≤ 0.5 µg/m, R > 1 µg/ml. Ofloxacin: S ≤ 0.25 µg/ml, R > 0.5 µg/ml. Ciprofloxacin: S ≤ 0.25 µg/ml, *R* > 0.5 µg/ml. Fold-change increases of MIC are shown in comparison to the QnrD-producing WT strain.

Similarly = the MICs increased for the *qnrD*-carrying *E. coli* exposed to sub-MIC of tobramycin as compared to growth in antibiotic-free medium (*Table 1*): 2- = 1.5- = and 2.7-fold higher for levofloxacin = ofloxacin and ciprofloxacin, respectively. These results showed that the aminoglycoside-induced SOS response increased quinolone (nalidixic acid and fluoroquinolones) MIC in line with the increased expression of *qnrD* in *E. coli*.

In *Enterobacterales*, high-level fluoroquinolone resistance is mainly due to the accumulation of mutations in the quinolone resistance-determining regions (QRDRs) of topoisomerase genes, the target of quinolones. As PMQRs lead only to low-level resistance, the emergence of clinically relevant fluoroquinolone resistance in strains harbouring PMQR has been attributed to the survival of a few isolates in the presence of fluoroquinolones that then develop QRDR mutations (*Martínez-Martínez et al., 1998*; *Robicsek et al., 2006*). However, in all these reports, the emergence of high-level fluoroquinolone resistance in enterobacterial isolates was directly linked to the specific selective pressure caused by the fluoroquinolones themselves.

To test the possibility that another class of antibiotics, the aminoglycosides (here, tobramycin), could allow the survival of *E. coli* isolates harbouring *qnrD*-plasmids, that further develop stepwise accumulation of QRDR mutations, we determined the mutant prevention concentrations (MPCs) of ciprofloxacin, levofloxacin and ofloxacin on a multi-susceptible clinical strain of *E. coli*, ATCC 25922, and its derivative harbouring pDIJ09-518a or pDIJ09-518aΔ*qnrD*. *E. coli* ATCC 25922 is the reference strain for antibiotic susceptibility testing recommended by the European and American committees on antimicrobial susceptibility testing and allowed us to compare MIC increases with standards. The MPC is the concentration that prevents the emergence of first-step resistant mutants within a susceptible population. When treating a patient, if the antibiotic concentration is below the MPC, the resistant mutant subpopulation within the wild-type population can emerge and potentially lead to therapeutic failure (*Cantón and Morosini, 2011*). These determinations were made with and without tobramycin pre-treatment (*Figure 6*). The presence of the plasmid without *qnrD* had no effect on mutant recovery whether or not these strains were exposed to a sub-MIC of tobramycin. No surviving mutants grew at 0.125, 0.06, and 0.06 µg/ml concentrations of ciprofloxacin, levofloxacin, or ofloxacin, respectively. When pDIJ09-518a was present however, concentrations of ciprofloxacin, levofloxacin, and ofloxacin that were 8- to 15.5-fold higher (1, 0.5, and 1 µg/ml, respectively) led to the recovery of over $10^4$, $10^6$, or $10^9$ *E. coli*/pDIJ09-518a colonies, respectively. Interestingly, it took concentrations as high as 2, 1, and 1.5 µg/ml of ciprofloxacin, levofloxacin, or ofloxacin, respectively, to abrogate the emergence of fluoroquinolone-resistant mutants in *E. coli*/pDIJ0-518a in the presence of tobramycin (*Figure 6*).

To further characterize resistance against ciprofloxacin, one of the main fluoroquinolones in current clinical use, we isolated five distinct colonies from a ciprofloxacin-containing plate of *E. coli* ATCC 25922/pDIJ09-518a incubated with tobramycin prior to performing the MPC assay. We determined the MICs of quinolone and sequenced the QRDR of the four-topoisomerase subunit-encoding genes (*gyrA*, *gyrB*, *parC*, and *parE*). The surviving mutants were all resistant or non-susceptible to nalidixic acid, ciprofloxacin, ofloxacin, and levofloxacin, and carried multiple mutations in QRDR (*Table 2*).

To bridge the gap between the aminoglycoside-induced SOS response and the increase of fluoroquinolones MPC of *E. coli*/pDIJ09-518a, we looked at the frequency of spontaneous mutations to rifampicin resistance for *E. coli* with or without pDIJ09-518a pre-exposed to sub-MIC of tobramycin (*Figure 5—figure supplement 1*). Upon pre-treatment to tobramycin, we found that the mutation frequency was 5.4-fold higher in *E. coli*/pDIJ09-518a compared to the plasmid-free *E. coli*. Overall, our results show that in *E. coli,* when carrying the *qnrD*-plasmid, sub-MICs of aminoglycosides potentiate the survival of *E. coli* isolates that would further develop QRDR mutations leading to higher fluoroquinolone resistance.

## Discussion

It is known that, conversely to *V. cholerae*, sub-MICs of aminoglycosides do not induce the SOS response in *E. coli*. In this study, however, we identified and characterized for the first time that the SOS response is induced in *E. coli* carrying a *qnrD*-plasmid upon exposure to aminoglycosides at a very low concentration (1% of MIC). This unexpected aminoglycoside-induced SOS response turns to be subsequent to NO accumulation in combination with aminoglycosides that eventually increase 8-oxo-G incorporation into DNA. Thereby, DNA damage appears through DSBs leading to induction to the SOS.

We found that two ORFs of the *qnrD*-plasmid, ORF3 and ORF4, were responsible for this SOS response induction by aminoglycosides in *E. coli*. Indeed, ORF3, which encodes a putative FAD-binding oxidoreductase, leads to NO production while ORF4, which encodes a putative CRP/FNR-like protein, inhibits *hmp* expression and thereby hampers the detoxification of NO. Although we did not evidence directly the 8-oxo-G incorporation hampering us to identify which targets aminoglycosides

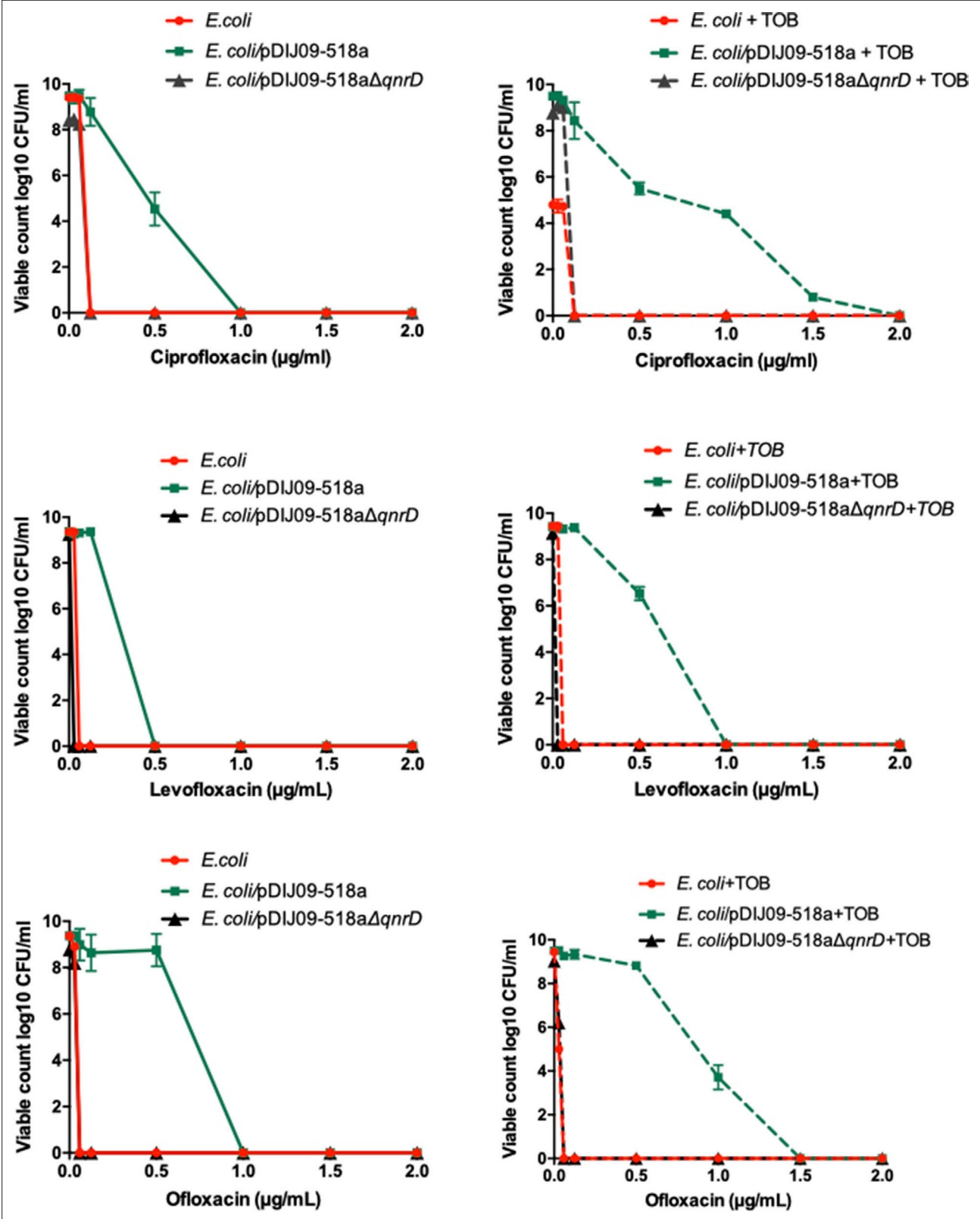

**Figure 6.** Aminoglycosides potentiate the selection of higher fluoroquinolone resistance in *E.coli* harbouring the small qnrD-plasmid. Mutant prevention concentrations (MPCs) of isogenic *E. coli* ATCC25922 strains with or without exposure to sub-MIC of tobramycin. *E. coli* ATCC25922 (red circle), *E. coli* ATCC25922/pDIJ09-518a (green square), and *E. coli* ATCC25922/pDIJ09-518aΔ*qnrD* (dark grey triangle).

*Figure 6 continued on next page*

*Figure 6 continued*

The online version of this article includes the following source data for figure 6:

**Source data 1.** CFU counting for MPC assay.

interact with, it has been previously reported that aminoglycosides and nitrosative stress caused oxidation of guanine (*Foti et al., 2012*; *Spek et al., 2001*). Taking this into account, we propose a model (*Figure 7*) of the pathway by which aminoglycosides induce the SOS response in *E. coli* carrying the *qnrD*-plasmid. In the absence of aminoglycosides, *E. coli* can repair the DNA damage resulting from the carriage of the *qnrD*-plasmid, and the SOS response is not induced (*Figure 7A*). In this case, the GO-repair system sanitizes efficiently the 8-oxo-G produced by the *qnrD*-plasmid-mediated NO accumulation. However, in the presence of aminoglycoside (*Figure 7B*), the level of oxidized guanine may increase. The resultant DNA damage yields genotoxic concentrations of alternate nucleotides that the GO-repair system can no longer sanitize efficiently, leading to induction of the SOS response. It is noteworthy that the burden caused by this *qnrD*-plasmid in *E. coli* upon exposure to tobramycin was not observed for *Providencia* spp. since no SOS induction was observed. We speculate that such a plasmid is harmless for *Providencia* spp. when exposed to aminoglycosides. Thereby, it could be one of the reasons why *qnrD*-plasmids may rarely be found in *E. coli*.

Our study shows that the widely accepted lack of SOS response induction in *E. coli* by aminoglycosides may not always be true. When strains bearing the *qnrD*-plasmid are exposed to sub-MICs of aminoglycosides, the SOS response does occur. This is a worrying issue, since we observed that the *qnrD*-plasmid is mobilizable (data not shown) and stable without antibiotic selective pressure. Selective pressures maintained by the overuse of antibiotics are the main drivers of resistance. In addition, sub-MICs of antibiotics can select for resistant bacteria and this occurs notably with fluoroquinolones (*Andersson and Hughes, 2017*). In this regard, expression of Qnr proteins and other PMQR normally confers only low levels of resistance to fluoroquinolones, but they were shown to significantly reduce the bactericidal activity of ciprofloxacin (*Allou et al., 2009*; *Guillard et al., 2013*) and to further facilitate selection for higher levels of resistance in Qnr-producing enterobacterial isolates exposed to fluoroquinolones, which can culminate in therapeutic failures (*Allou et al., 2009*; *Guillard et al., 2013*; *Martínez-Martínez et al., 1998*; *Robicsek et al., 2006*). In addition, it has also recently been shown

**Table 2.** Quinolone resistance-determining region (QRDR) mutations and minimal inhibitory concentration of quinolones for surviving mutants obtained in the mutant prevention concentration (MPC) assay.

| Strain* | Mutant | MPC (µg/ml) | QRDR mutations | | | | MIC (µg/ml)[†] | | | | | | | |
|---|---|---|---|---|---|---|---|---|---|---|---|---|---|---|
| | | | GyrA | GyrB | ParC | ParE | NAL | | CIP | | OFX | | LVX | |
| *E. coli* ATCC25922/ pDIJ09-518a | #1 | 1 | S83L | - | - | - | >256 | R | 0.125 | S | 0.38 | I | 0.19 | S |
| | #2 | 1 | S83W | - | - | - | 24 | R | 0.25 | S | 0.5 | I | 0.38 | S |
| | #3 | 1 | S83W | - | - | - | >256 | R | 0.38 | I | 0.75 | R | 0.38 | S |
| | #4 | 1 | S83W | - | - | - | 24 | R | 0.19 | S | 0.5 | I | 0.25 | S |
| | #5 | 1 | S83W | - | - | - | >256 | R | 0.125 | S | 0.38 | I | 0.19 | S |
| *E. coli* ATCC25922/ pDIJ09-518a + TOB | #1 | 2 | S83W | - | - | - | >256 | R | 1 | R | 4 | R | 1 | I |
| | #2 | 2 | S83W | - | G78D | - | >256 | R | 0.38 | I | 2 | R | 0.75 | I |
| | #3 | 2 | S83W | - | G78D | - | >256 | R | 1.5 | R | 4 | R | 0.75 | I |
| | #4 | 2 | S83W | - | G78D | - | >256 | R | 1 | R | 6 | R | 1 | I |
| | #5 | 2 | S83W | - | - | - | >256 | R | 1 | R | 4 | R | 1 | I |

*+TOB stands for strains exposed to sub-MIC of tobramycin prior to MPC assay.

[†]Susceptibility testing categories according to EUCAST clinical breakpoints. Nalidixic acid: R > 16 µg/ml. Levofloxacin: S ≤ 0.5 µg/ml, R > 1 µg/ml. Ofloxacin: S ≤ 0.25 µg/ml, R > 0.5 µg/ml. Ciprofloxacin: S ≤ 0.25 µg/ml, R > 0.5 µg/ml.

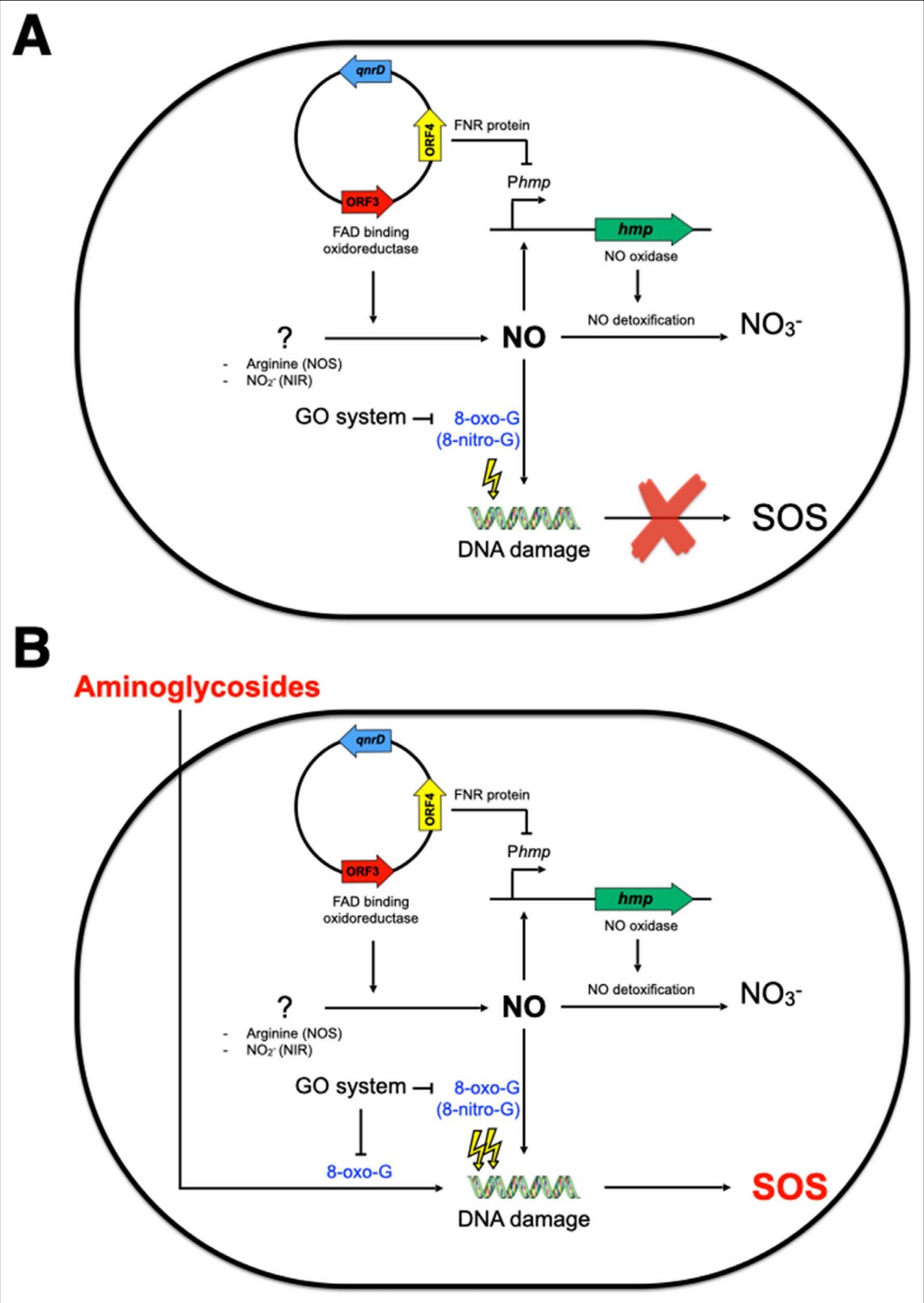

**Figure 7.** Model of SOS response induction by aminoglycosides in *E. coli* bearing the small *qnrD*-plasmid. Schematic representation of the network leading to SOS induction in *E. coli*/pDIJ09-518a when not (**A**) or exposed (**B**) to sub-minimum inhibitory concentration (MIC) of aminoglycosides. NOS, nitric oxide species; NIR, nitrite reductase.

that *qnrB* promotes DNA mutations and thereby fluoroquinolone-resistant mutants by triggering DNA replication stress (*Li et al., 2019*).

In this context, the SOS regulation of such PMQR genes have clinical implications, not only in terms of infectious disease treatments, but also to prevent the dissemination of resistance genes. It has been clearly demonstrated that the *qnrB*-mediated quinolone resistance is induced upon exposure to sub-MICs of fluoroquinolone (*Da Re et al., 2009*). Therefore, describing for the first-time aminoglycosides that induce the SOS response in *E. coli* carrying a low level of fluoroquinolone resistance determinant (*qnrD*-plasmid) could have worthwhile therapeutic implications by increasing the odds of mutations during the SOS induction, since each of these classes of antibiotics is commonly used as first- or second-line treatment.

## Materials and methods

### Bacterial strains, plasmids, primers, and growth conditions

The bacterial strains, plasmid constructs, and primers for PCR analysis used in this work are shown in *Supplementary files 3 and 4* (*Blattner et al., 1997*; *Mount et al., 1972*). The *E. coli* Δ*hmp* (KEIO collection) (*Baba et al., 2006*) was graciously provided by J.-M. Ghigo (Pasteur Institute, Paris). This allele was transduced in MG1656, using P1 phage and selected on agar plates with 50 µg kanamycin/ml. Experiments were performed in LB or in minimum medium at 37°C. For genetic selections, antibiotics were added to media at the following concentrations: 100 µg ampicillin/ml, 0.03 and 0.06 µg ciprofloxacin/ml, 50 µg kanamycin/ml, 50 µg streptomycin/ml, or 100 amoxicillin µg/ml. For each strain, the MIC of antibiotics was determined twice for each antimicrobial agent using *E*-test strips (bioMérieux, Marcy l'Etoile, France). The sub-MICs (e.g. 1% of MIC) of specified antibiotics were used to induce the SOS response, as follows (final concentrations [µg/ml]): ciprofloxacin (CIP) 0.06, gentamicin (GM) 0.00125, mitomycin C (MMC) 0.1, and tobramycin (TM) 0.001.

### WT::*qnrD* and WT::pDIJ09-518a strains construction

DNA fragments were generated by PCR in order to amplify the *qnrD* gene with its own promoter or the pDIJ09-518a from the native plasmid and the chromosomal *cynX* and *lacA* intergenic region from the MG1656 genomic DNA. The three PCR fragments were digested with DpnI and purified with a Qiagen PCR Purification Kit (Qiagen). The three PCR products were assembled as one large fragment (5′ *cynX* – insert – *lacA*3′) by Gibson Assembly (New England Biolabs). The assembled DNA was transformed into electrocompetent WT *E. coli* where they were recombined into the *E. coli* WT genome. The transformed bacteria were selected using 0.06 µg ciprofloxacin/ml by incubation for 24 hr, at 37°C. The insertion was verified by colony PCR with four primers pairs: AB09/AB06 (*qnrD*), AB13/AB08 (plasmid), AB06/AB12 (*qnrD*), and AB07/AB16 (plasmid) (*Supplementary file 4*). The positive clones were verified by sequencing using AB09/AB12 and AB13/AB16.

### Plasmid constructions

The *recA*, *mutT*, and *hmp* genes with their own promoters were amplified from the *E. coli* WT genome, with the corresponding Forward/Reverse primers shown in *Supplementary file 4*. The PCR products were purified and cloned into pCR2.1 TOPO, hereafter called pΦ (Thermo Fisher Scientific, France) to generate pRecA, pMutT, and pHmp, respectively, and selected on plates containing 50 µg kanamycin/ml. The same protocol was used for the complementation of the ORF3 and ORF4 genes to generate pORF3 and pORF4. However, these ORFs were amplified from *E. coli*/pDIJ09-518a.

pDIJ09-518aΔ*qnrD* was obtained by PCR amplification of the native pDIJ09-518a plasmid as DNA template excluding the *qnrD* gene, using the primers described in *Supplementary file 4*. The primers were obtained using the NEBuilder Assembly Tool (New England Biolabs). After digestion by DpnI (New England Biolabs) and purification of PCR products (Qiagen), the fragment obtained was transformed into electrocompetent *E. coli* WT or WT::*qnrD*. Transformants were selected on agar plates containing 0.06 µg ciprofloxacin/ml and were analysed by PCR as described above. The same protocol was used to obtain pDIJ09-518aΔORF3, -ΔORF4, and -ΔORF3ΔORF4, using the indicated primers (*Supplementary file 4*).

pDIJ09-518a LexA-box* substitution of CGT to AGC, in the LexA-box or SOS-box-binding site, was obtained by using a modified Quick-Change II Site-Directed Mutagenesis kit (Agilent). Briefly, we used

the primers containing the substitution (*Supplementary file 4*), as described by the manufacturer. The elongation temperature used for the amplification was modified to 68°C for 2 min. Eighteen cycles of amplification were sufficient to obtain a proper amount of modified DNA template. After digestion by DpnI and purification of PCR products, the fragment obtained was transformed into *E. coli* competent TOP10 cells (Thermo Fisher). Transformants were selected on agar plates containing 0.06 µg ciprofloxacin/ml and analysed by PCR as described above.

## DNA manipulation and genetic techniques

Genomic DNA was extracted and purified using the Qiagen DNeasy purification kit (Qiagen, Courtaboeuf, France). Isolation of plasmid DNA was carried out using the QIAprep Spin Miniprep kit (Qiagen). Gel extractions and purifications of PCR products were performed using the QIAquick Gel Extraction kit (Qiagen) and QIAquick PCR Purification kit (Qiagen). PCR verifying experiments were performed with Go Taq Green Master Mix (Promega, Charbonnières les Bains, France), and PCRs requiring proofreading were performed with the Q5 High-Fidelity DNA Polymerase (New England BioLabs, Evry, France) as described by the manufacturers. Restriction endonucleases DpnI was used per the manufacturer's specifications (New England BioLabs). All DNA manipulations were checked by DNA sequencing (GENEWIZ, Takeley, England).

## RNA extraction and qRT-PCR

Strains were grown in LB at 37°C with shaking to exponential phase ($OD_{600}$ = 0.5–0.7). Six biological replicates were prepared, if not specify otherwise in the legends. One percent of the MICs of indicated antibiotics was then been added to the culture for 30 min to allow for induction of the SOS response. One culture was kept as an antibiotic-free control. Five hundred microliters of exponentially growing cells were stabilized in 1 ml of RNAprotect Bacteria Reagent (Qiagen). After treatment of the culture pellet with lysozyme (Qiagen), subsequent RNA extractions were performed using the RNeasy Mini Kit (Qiagen). The genomic DNA contaminating the samples was removed with TURBO DNA-free Kit (Ambion, Thermo Fisher Scientifics) at 37°C, for 30 min. First-strand cDNA synthesis and quantitative real-time PCR were performed with KAPA SYBR FAST (CliniSciences, Nanterre, France) on the LightCycler 480 (Roche Diagnostics, Meylan, France) using the primers indicated in *Supplementary file 4*. Transcript levels of each gene were normalized to *dxs* as the reference gene control. Gene expression levels were determined using the $2^{-\Delta\Delta Cq}$ method (*Bustin et al., 2009*; *Livak and Schmittgen, 2001*) in respect to the MIQE guidelines. Relative fold-difference was expressed either by reference to antibiotic-free culture or the WT strain in LB. All experiments were performed as six independent replicates (if not specify otherwise) with all samples tested in triplicate. Cq values of technical replicates were averaged for each biological replicate allowing us to obtain the ΔCq. After exponential transformation of the ΔCq for the studied and the normalized condition, medians and upper/lower values were determined.

## Flow cytometry

Flow cytometry experiments were performed as described (*Baharoglu et al., 2014*; *Baharoglu et al., 2013*; *Baharoglu et al., 2010*) and repeated at least three times on overnight cultures in MH or MH+ sub-MIC tobramycin (0.001 µg/ml). Briefly, overnight cultures of *E. coli* MG1655 and *E. coli* MG1655/pDIJ09-518a and their derivatives were prepared in LB broth with or without sub-MIC of tobramycin and diluted 40-fold into (phosphate-buffered saline Invitrogen). The GFP fluorescence was measured using the Miltenyi MACSQuant device.

## Detection of intracellular ROS by DHR-123 and NO by DAF-2-DA

Overnight cultures of *E. coli* WT and *E. coli* WT/pDIJ09-518a and its derivatives were diluted 100-fold in fresh LB broth. DHR-123 (Sigma) or DAF-2 DA (Sigma) were added to 5 ml LB to a final concentration of $2.5 \times 10^{-3}$ or 10 µM, respectively. Sub-MICs of ciprofloxacin or tobramycin were added for 30 min to the cultures at $OD_{600}$ 0.5–0.7, and when stated, cPTIO (Enzo Life Sciences) used at the final concentration of 5 µM. Two hundred microliters of bacterial cultures were then added to 96-well black flat-bottom plates (Corning) in triplicate. The DHR-123 fluorescence was measured at 507/529 nm (excitation/emission wavelength) whereas the DAF-2 DA fluorescence was measured at 491/513 nm,

on a SAFAS Xenius XC (Safas, Monaco). The values used were corrected by subtracting the values from the negative controls. Experiments were repeated at least three times in triplicates.

### Detection of intracellular NO by flow cytometry

Overnight cultures of *E. coli* MG1655 and *E. coli* MG1655/pDIJ09-518a were diluted 100-fold into fresh LB broth, and growth until an $OD_{600nm}$ of 0.2 was reached. The DAF-2A, the NO scavenger cPTIO (5 µM final concentration), and sub-MIC of tobramycin were then added for 30 min to the cultures. Flow cytometry experiments were performed as described (*Baharoglu et al., 2010*) and repeated at least five times. DAF-2A fluorescence was measured using the Miltenyi MACSQuant device.

### MIC determination

MICs were determined by *E*-test (bioMérieux, Marcy l'Etoile, France) in accordance with EUCAST guidelines. Briefly, MICs of nalidixic acid, ciprofloxacin, levofloxacin, and ofloxacin were determined on MH agar inoculated with a 0.5 McFarland (~$10^8$ colony-forming unit [CFU/ml]) bacterial suspension. MICs were read after incubation for 18 hr at 37°C.

### Growth curves

Overnight cultures of *E. coli* WT and *E. coli* WT/pDIJ09-518a were diluted 100-fold into LB broth. Growth curves were obtained using an automated turbidimetric system (Bioscreen C, LabSystem) at 37°C with shaking during 24 hr. Optical density measurements at 600 nm ($OD_{600}$) were performed every 5 min with 10 s of shaking prior to reading. The $OD_{600}$, corrected with values from the negative controls, and the corresponding Log10 CFU/ml, were used to fit the growth curves of each studied strain. For both strains, biological experiments were performed in triplicate.

### Plasmid stability

*E. coli* carrying the pDIJ09-518a was inoculate in a LB non-selective medium at 37°C and 200 rpm, for 30 days. Every 24 hr of growing, an aliquot of 100 µl was inoculated in fresh LB non-selective medium. At days 5, 10, 15, 20, 25, and 30, an aliquot of 100 µl was spread on a LB agar non-selective plates (to measure the living cells) and another 100 µl on a LB agar selective medium plates (here CIP to measure those yet harbouring the plasmid) and incubate for 24 hr at 37°C before proceeding to a PCR colony of five independent colonies. Biological experiments were performed in triplicate.

### Viability study after tobramycin treatment

Overnight cultures of WT or WT/pDIJ09-518a or their derivatives strains were diluted 100-fold into LB broth and LB broth supplemented with 0.001 µg/ml tobramycin and growth 37°C for 24 hr. Aliquots were plated on LB agar and incubated at 37°C for 24 hr. The CFUs/ml were counted. The bars represent the percentage of CFU counted after tobramycin treatment over LB. For all strains, biological experiments were performed in duplicate.

### Spontaneous mutation frequency after treatment with sub-MIC of tobramycin

Twelve single colonies of each strain (for each strain, six colonies in two independent assays) were grown overnight in LB supplemented with sub-MIC of tobramycin (0.001 µg/ml). Appropriate dilutions were plated on LB, and 1 ml of culture was centrifuged and plated on 200 µg/ml rifampin plates. The frequencies of spontaneous mutations to rifampicin resistance correspond to the rifampin-resistant CFU count over the total number of CFUs.

### Bioinformatic analysis and statistics

The prevalence and the dissemination of the small *qnrD*-plasmids into *Enterobacterales* were analysed on the National Centre for Biotechnology Information (NCBI) database. The small *qnrD*-plasmids were classified into three groups (*Morganellaceae*, *E. coli*, and others *Enterobacterales*) and into two classes, considering the length of known *qnrD*-plasmids (pDIJ09-581a or p2007057-like).

The LexA-box consensus sequence logo was established using WebLogo http://weblogo.berkeley.edu/logo.cgi (*Crooks et al., 2004*) taking into account the 16nt of SOS-box of all 53 fully sequenced *qnrD*-plasmids.

For qRT-PCR, a Wilcoxon matched-pairs signed-rank test was used to compare median of fold changes (*Livak and Schmittgen, 2001*; *Schmittgen and Livak, 2008*; *Yuan et al., 2006*).

For ROS and NO detection, a two-way ANOVA with ad hoc tests (Tukey's multiple comparisons test or a Dunn's multiple comparisons test) was used to compare the measured values between the different strains and conditions.

For spontaneous mutations rates, a Wilcoxon matched-pairs signed-rank test was used to compare median of Rif CFU/ml/total CFUs counted.

All the tests were performed using GraphPad Prism version 6. Degree of significance is indicated as follows: $*p < 0.05$; $**p < 0.01$; $***p < 0.001$; $****p < 0.0001$.

## Acknowledgements

We thank Dr Valerian Dormoy for his careful reading of the manuscript and scientific discussion of the results, Dr Christine Terryn of PICT-URCA platform for technical assistance in imaging core facilities, Dr Arnaud Bonnomet from Inserm UMR-S 1250 for their technical assistance, and Prof Grace Stockton-Bliard for proofreading the manuscript. Funding: This work was supported by the Université de Reims Champagne-Ardenne [to AB, TG, and CDC] and the Conseil Régional de Champagne-Ardenne, the Association pour le Développement de la Microbiologie et de l'Immunologie Rémoises, and the International Union of Biochemistry and Molecular Biology [to AB]. Work in the Mazel lab work was supported by the French Government's Investissement d'Avenir program Laboratoire d'Excellence 'Integrative Biology of Emerging Infectious Diseases' [ANR-10-LABX-62- IBEID], by Institut Pasteur and by CNRS.

## Additional information

### Funding

| Funder | Grant reference number | Author |
|---|---|---|
| Université de Reims Champagne-Ardenne | | Anamaria Babosan Christophe de Champs Thomas Guillard |
| Conseil Régional de Champagne-Ardenne | | Anamaria Babosan Christophe de Champs Thomas Guillard |
| Association pour le Developpement de la Microbiologie et de l'Immunologie Rémoises | | Anamaria Babosan |
| International Union of Biochemistry and Molecular Biology | | Anamaria Babosan |
| Agence Nationale de la Recherche | ANR-10-LABX-62- IBEID | Didier Mazel |
| Centre National de la Recherche Scientifique | | Zeynep Baharoglu Didier Mazel |
| Institut Pasteur | | Zeynep Baharoglu Didier Mazel |

The funders had no role in study design, data collection, and interpretation, or the decision to submit the work for publication.

### Author contributions

Anamaria Babosan, Conceptualization, Data curation, Formal analysis, Investigation, Methodology, Resources, Validation, Visualization, Writing - original draft, Writing – review and editing; David Skurnik, Conceptualization, Writing - original draft, Writing – review and editing; Anaëlle Muggeo, Investigation, Resources, Writing – review and editing; Gerald B Pier, Marie-Cécile Ploy, Resources,

Writing – review and editing; Zeynep Baharoglu, Sophie Griveau, Fethi Bedioui, Investigation, Writing – review and editing; Thomas Jové, Investigation, Resources, Validation; Sébastien Vergnolle, Sophie Moussalih, Investigation; Christophe de Champs, Conceptualization, Validation, Writing – review and editing; Didier Mazel, Conceptualization, Funding acquisition, Resources, Writing - original draft, Writing – review and editing; Thomas Guillard, Conceptualization, Formal analysis, Funding acquisition, Methodology, Project administration, Resources, Supervision, Validation, Writing - original draft, Writing – review and editing

Author ORCIDs
Gerald B Pier http://orcid.org/0000-0002-9112-2331
Zeynep Baharoglu http://orcid.org/0000-0003-3477-2685
Didier Mazel http://orcid.org/0000-0001-6482-6002
Thomas Guillard http://orcid.org/0000-0002-3795-0398

Decision letter and Author response
Decision letter https://doi.org/10.7554/eLife.69511.sa1
Author response https://doi.org/10.7554/eLife.69511.sa2

## Additional files

### Supplementary files
- Transparent reporting form
- Supplementary file 1. Genetic background of qnrD positive enterobacterial isolates.
- Supplementary file 2. Minimum-inhibitory concentrations for antibiotics used as SOS-response inducers.
- Supplementary file 3. Strains and plasmids.
- Supplementary file 4. Primers used for this study.

### Data availability
Source data are provided as a Source Data file. Flow cytometry data have been deposited in FlowRepository as FR-FCM-Z3MR repository ID (http://flowrepository.org/id/RvFrzhOtiB4Hrd9yMMTEF2gAckZvYVa365phD9U0fVTabQb7ibCDqV8Gzbgb02dm).

The following dataset was generated:

| Author(s) | Year | Dataset title | Dataset URL | Database and Identifier |
|---|---|---|---|---|
| Guillard T | 2021 | DAF-2A fluorescence quantification | http://flowrepository.org/id/RvFrzhOtiB4Hrd9yMMTEF2gAckZvYVa365phD9U0fVTabQb7ibCDqV8Gzbgb02dm | FlowRepository, FR-FCM-Z3MR |

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
