## [Editor Report]

This manuscript describes how a small plasmid containing a quinolone resistance determinant changes the cellular response to sub-inhibitory concentrations of Tobramycin. The authors report that *E. coli* cells carrying this plasmid undergo nitrosative stress mediated by two previously uncharacterized genes, which results in induction of the SOS response. These findings are interesting and relevant for readers across microbiology and genetics fields.

---

## [Decision Letter]

**Decision letter after peer review:**

Thank you for submitting your article "A qnr-plasmid allows aminoglycosides to induce SOS in *Escherichia coli*" for consideration by *eLife*. Your article has been reviewed by 3 peer reviewers, and the evaluation has been overseen by Bavesh Kana as the Senior and Reviewing Editor. The following individual involved in review of your submission has agreed to reveal their identity: Xian-Zhi Li (Reviewer #1).

Essential revisions:

Main Concerns:

1. The major weakness of the work is that it does not assess whether the SOS induction by AGs leads to increased mutagenesis as speculated.

2. The qnrD plasmid containing ORFs 3 and 4 is not commonly found in *E. coli* (Table S3 and Figure S1). It is adequate that most of the work has been performed in this genetically tractable organism. Nevertheless, it would be important to assess whether the same AG-mediated SOS induction occurs in bacteria that usually carry this type of plasmid (P. mirabilis or other Morganellaceae). This is a simple qRT PCR experiment which could add a lot to the significance of the paper. Perhaps the nitrosative stress (which is obviously a deleterious effect) is particular to *E. coli* and that could be one of the reasons why this particular plasmid architecture occur rarely (or never?) in this species.

3. There is no direct evidence provided for increase in 8-oxo-G incorporation. Thus, discussion should carefully be worded (Line 425). Indeed, the target(s) (molecules) with which aminoglycoside interact with as inducers remain unknown, although the aminoglycoside effect is dependent on the GO repair system and indirectly results in DNA damage (Figure 3). It is difficult to extend this to the description that aminoglycosides and others form 8-oxo-G (in L280-282). If any, the aminoglycoside effect is indirect. Clarity is needed.

4. Further to the above, the qnrD expression was induced by aminoglycosides (L130). However, it remains unknown whether the elevated 2.8-fold increase in qnrD expression in the presence of tobramycin could produce an impact on fluoroquinolone resistance phenotype. First, an aminoglycoside at 1% MIC is expected to exhibit no antibacterial activity. Thus, a simple fluoroquinolone MIC testing can be done in the presence and the absence of very low levels of aminoglycosides that induce SOS response. A reduced fluoroquinolone susceptibility would argue in favor of author-suggested significance of SOS induction findings.

5. As indicated by authors (L414-415), the functional roles of ORF3 and ORF4 via biochemical studies were not directly measured. Instead, sfiA expression and fluorescence indicators were assessed. This represents a weakness of the study. Can the authors address this?

6. How do the experiments in figure S2 address plasmid stability during growth by measuring OD? Plasmid stability in the experimental conditions is an important issue.

7. Comparing Figures 3B and 1E, it can be observed that deletion of both recF and recB have significant effects on SOS induction by AG in cells carrying the qnrD plasmid. These results are very important for the model and a qRT measurement of sulA in recF and recB mutants would be an important confirmation for the results obtained with plasmid based assays, which could be influenced by the reporter plasmid copy number and stability in the different genotypes analyzed.

8. The complementation of ORF 3 and 4 mutants (Figures 4A and 4C) is incomplete, the same SOS induction level as observed in the wt is never reached. Could this be due to the deletion of one ORF influencing on the expression of the other? Additionally, a more appropriate comparison with cells carrying the empty expression vector may help to solve this issue.

9. In contrast with the detailed analysis described in the initial sections, the information provided on the products of ORF3 and ORF4 is scanty. This is unfortunate as these products and their activities are the main findings of the story. Hence, more detailed characterization of these putative proteins seems advisable to support the authors's model. Bioinformatic analysis should be shown in more detail, the existence of the predicted proteins should be proven and the putative DNA binding activity of the ORF4 product should be shown. Otherwise, the possibility of indirect effect will remain open.

Other Comments that should be addressed:

1. The study introduces two groups of qnrD plasmids (L77). Do the authors have information about their copy numbers? For example, what is the copy number of pDIJ09-518a? This information will assist in understanding the basal expression of qnrD and the other genes in the absence of an inducer, and thus the significance of aminoglycoside induction of SOS response.

2. BLAST was carried out to find homologs of ORF3 and ORF4 (L366-373). Please include gene/product access information for an Erythrobacter spp protein and an O2-responsive regulator from Pedobacter panaciterrae. Are there any reference(s) to support these proteins that are used for comparison?

3. L326. In the absence of RecB, the SOS response was somehow (1.4-fold change; but seems statistically significant [Figure 3B]) observed in the presence of tobramycin, suggesting the possible presence of a RecF-independent SOS response pathway. For comparison, the control for both recF and recB, the ratios are quite below 1, Hence, the actual changes are larger. Suggest additional discussion. (Additionally, "LB" is indicated in text while Figure 3B shows "MH" – please clarify.)

4. L65. Among 53 fully sequenced qnrD plasmids, do they share the same orf3 and orf4 reported in this manuscript? In other word, how often are these ORFs observed in qnrD plasmids?

5. L221. Is the description "However, the underlying mechanism explaining these findings has yet to be identified" simply due to QnrD protection of ciprofloxacin target?

6. L402-403. The description "We found that ORF4 did not alleviate the SOS response after exposure to aminoglycoside sub-MICs as measured by sfiA expression (Figure 4C)" is not reflective of Figure 4C because orf3 deletion reduced sifA expression (statistically significant per Figure 4C legend). Please verify. As well, obviously, the complementation with pORF4 did not work in Figure 4C.

7. L431. The proposed Model (should be Figure 5, not Figure 4) is considered to reflect the findings. However, the wording should be careful. For example, in L436, the description "the level of oxidized guanine increases" may be revised as "the level of oxidized guanine may increase" as no measurement for 8-oxo-G was done in this study.

8. In contrast with the detailed analysis described in the initial sections, the information provided on the products of ORF3 and ORF4 is scanty. This is unfortunate as these products and their activities are the main findings of the story. Hence, more detailed characterization of these putative proteins seems advisable to support the authors's model. Bioinformatic analysis should be shown in more detail, the existence of the predicted proteins should be proven and the putative DNA binding activity of the ORF4 product should be shown. Otherwise, the possibility of indirect effect will remain open.

9. Absolute numbers of β-gal activities should be provided when lac fusions are used. This would help the reader to assess the magnitude of the phenomena described in the manuscript.

10. Please revise the statement that you proved functionality of the SOS box in the qnr locus of the plasmid family under study. You did so in the archetypal plasmid pDIJ109-518a. The writing may however suggest that functionality was proven in the entire plasmid collection provided in Table S3.

*Reviewer 2 (Recommendations for the authors):*

1. Some of the data files (spreadsheets) have French words/phrases. If they are intended to be published alongside the paper, these terms should be translated to English.

2. Figure S1 and introduction. Are ORFs 3 and 4 from the "small" qnr plasmids studied in this work also present in the "large" qnr plasmid? This is not clear from the text or Figure S1. As an alternative, Table S3 could indicate if ORFs 3 and 4 homologues are present in each of the qnrD plasmids.

3. This is a matter of choice, but sulA is a more widely use gene name than sfiA, and should be used throughout the text. sfiA is the original name, but modern gene annotations in bacterial genomes use sulA.

4. Lines 118-119: Revise this information. In figure 1A, the consensus observed upstream to qnrD actually has 15 out of 16 bases identical to the *E. coli* consensus shown.

5. Lines 291 to 293. Phrasing is not good, suggest: the SOS response induction could result from the deleterious effects…

6. Lines 402-403: "We found that ORF4 did not alleviate the SOS response after exposure to aminoglycoside sub-MICs, as measured by sfiA expression". In my view the SOS response is attenuated, although not completely, after ORF4 deletion. Maybe a more precise statement would be: "We found that ORF4 deletion does not completely abolish SOS induction by aminoglycoside sub-MICs, as measured by sfiA expression".

7. There is an unnecessary repetition of results in Figures 4A, 4C and 4E (wt, wt/pDIJ09). All these results could be merged into a single graph.

8. Data from Figure 4B and the model depicted in Figure 5 implicate that ORF4 can act on its own (independently from ORF3 or stressors) as a negative regulator of hmp, contributing to nitrosative stress. Nevertheless, if this was the case, I would expect the following in qRT-PCR experiments shown in Figure 4D:

Wt/pDIJ09 – lower hmp levels than cells without plasmid, due to (ORF4) action.

Wt/pDIJ09 del ORF4 – same hmp levels as cells without plasmid. No negative regulator present in spite of the rest of plasmid.

Wt/pDIJ09 del ORF4/pORF4 – lower hmp levels than cells without plasmid, due to in trans negative regulator (ORF4) action.

Can the authors perhaps add some comment? Is it possible that there is another interpretation for these data?

9. Lines 622-623 mention a mutation rate analysis, but such experiments are not shown in the paper.

10. Viability after tobramycin treatment (lines 601-606): It is not clear for how long cells were exposed to the antimicrobial.

11. Material and Methods or Figure legends should be more clear regarding the concentration of antimicrobials used in each experiment. For example, when comparing *E. coli* cells carrying or not the qnrD plasmid, each strain was treated with different sub-MIC concentrations of cipro to account for their different resistance levels? In Materials and methods (lines 479-481), 1% of the MIC of Cipro (0.06 µg/mL) is indicated as the sub-MIC concentration used. However, Table S1 indicates a MIC of 0.004 µg/mL for the wt strain, and 0.094 µg/mL for the strain carrying the qnr plasmid. This information is confusing and should be revised.

12. Were experiments measuring SOS induction with the GFP reporter performed in LB or MH medium? Material and Methods (lines 559-561) is contradictory on this regard. Figure legends indicate LB, but the Y axis of figures 1E and 3B show MH.

13. Line 588 mention levofloxacin and ofloxacin, but these antimicrobials were not used in this work.

*Reviewer 3 (Recommendations for the authors):*

1. The authors use the term "aminoglycosides" in multiple sections of the manuscript, including the title, the abstract and the discussion. However, only one aminoglycoside (tobramycin) has been used throughout the study. Gentamicin was used at the beginning of the study but disappears at some point. Please clarify.

[Editors' note: further revisions were suggested prior to acceptance, as described below.]

Thank you for submitting your article "A qnr-plasmid allows aminoglycosides to induce SOS in *Escherichia coli*" for consideration by *eLife*. Your article has been reviewed by 3 peer reviewers, and the evaluation has been overseen by Bavesh Kana as the Senior and Reviewing Editor. The following individual involved in review of your submission has agreed to reveal their identity: Xian-Zhi Li (Reviewer #1).

Essential revisions:

Whilst reviewers concurred that your manuscript has improved, significant concerns around complementation still remain. Please address these. Other points are indicated below.

1. The manuscript is somewhat confusing regarding Figure and supplemental figure names. It is often difficult to understand which figured is being commented in the text (Examples: line 228, line 283, line 372). Also, supplemental figures are not numbered in the final document, making it hard to follow the text.

2. Page 8 lines 187-191. This result deserves a better discussion in the paper. The burden presented by this plasmid for *E. coli* is not present in Providencia. Therefore, this may one of the reasons why qnrD plasmids are rarely found in *E. coli*. SOS induction per se is a symptom of a stressed cell in a disadvantageous situation. Therefore, lack of SOS induction for Providencia means that this species is not harmed by presence of the plasmid, whereas the opposite is true for *E. coli*.

3. The plasmid used for complementation in experiments shown in Figure 4A clearly affects SOS induction by Tobramycin in cells carrying pDIJ09 (second versus fourth brown bar in figure 4A). This complicates the interpretation of all complementation experiments and should be revisited and the discussion should carefully reflect this limitation. There is no reference for the pTOPO vector used for complementation. Does it carry kanamycin or another aminoglycoside resistance? Could some sort of low-level cross resistance conferred by the plasmid be interfering with the effects of Tobramycin? Perhaps complementation with a chromosomal integration of ORFs 3 and 4 as done in other parts of the paper would be better.

4. In figure 3D, why does deletion of hmp does not increase sulA induction after Tobra treatment? The result does not make sense when compared to the overexpression and should be discussed in further detail.

5. Lines 482 to 490 are very confusing and need re-writing. The MPC experiments need a detailed description in the methods section.

6. Mutagenesis data should ideally represent at least 3 experiments. Authors should also describe in more detail how these experiments were performed (initial inoculum, hours of growth in the presence of the drug, etc).

7. The titles of Tables 1 and 2 includes "fluoroquinolone", while nalidixic acid is not a fluoroquinolone. A footnote could be helpful. In Abstract, please write "encodes", not "encode" (L47) and write "codes", not "code" (L48).

8. As ORF3 and ORF4 have not been characterized, it should be made clear that the identities of the ORF3 and ORF4 products are putative.

*Reviewer #1 (Recommendations for the authors):*

This is a revised manuscript. The authors have conducted additional experiments to address the major concerns from the previous reviewers. The addition of Tables 1 and 2/Figure 5 has enhanced the manuscript. The authors' response is considered satisfactory. There are no major comments from this reviewer.

*Reviewer #2 (Recommendations for the authors):*

This version of the manuscript has addressed many of the previous points raised by the reviewers. The work is interesting, but there are still some issues that need to be addressed:

For some reason, this version of the paper is very confusing regarding Figure and supplemental figure names. It is often difficult to understand which figured is being commented in the text (Examples: line 228, line 283, line 372). Also, supplemental figures are not numbered in the final document, making it hard to follow the text.

Page 8 lines 187-191. In my opinion, this result deserves a better discussion in the paper. The burden presented by this plasmid for *E. coli* is not present in Providencia. Therefore, this may one of the reasons why qnrD plasmids are rarely found in *E. coli*. SOS induction per se is a symptom of a stressed cell in a disadvantageous situation. Therefore, lack of SOS induction for Providencia means that this species is not harmed by presence of the plasmid, whereas the opposite is true for *E. coli*.

The plasmid used for complementation in experiments shown in Figure 4A clearly affects SOS induction by Tobramycin in cells carrying pDIJ09 (second versus fourth brown bar in figure 4A). This complicates the interpretation of all complementation experiments and should be revisited. There is no reference for the pTOPO vector used for complementation. Does it carry kanamycin or another aminoglycoside resistance? Could some sort of low-level cross resistance conferred by the plasmid be interfering with the effects of Tobramycin? Perhaps complementation with a chromosomal integration of ORFs 3 and 4 as done in other parts of the paper would be better.

In figure 3D, why deletion of hmp does not increase sulA induction after Tobra treatment? The result does not make sense when compared to the overexpression and should be better discussed.

Lines 482 to 490 are very confusing and need re-writing. The MPC experiments need a detailed description in the methods section.

Mutagenesis data should ideally represent at least 3 experiments. Authors should also describe in more detail how these experiments were performed (initial inoculum, hours of growth in the presence of the drug, etc).

*Reviewer #3 (Recommendations for the authors):*

The authors have done a good job responding to the reviewers' suggestions. Relevant points have been clarified and risky statements have been toned down in the revised version. The story remains incomplete as ORF3 and ORF4 have not been characterized. As a consequence, a crucial point in the story (the mechanism that triggers NO accumulation) remains enigmatic. Despite this shortage, I agree with the authors that qnr-mediated induction of the SOS response by aminoglycosides is an interesting phenomenon. My only suggestion at this point is to modify the text wherever pertinent (e. g., lines 47-49 and 388-396) to make it clear that the identities of the ORF3 and ORF4 products are putative.

---

## [Author Response]

Essential revisions:1. The major weakness of the work is that it does not assess whether the SOS induction by AGs leads to increased mutagenesis as speculated.

In the revised manuscript, we added the results of the spontaneous mutation ratio we obtained upon pre-treatment with sub-MIC of tobramycin. These results are mentioned lines 503-510 and shown in Figure S6. We found that the mutation frequency was 5.4-fold higher in *E. coli*/pDIJ09-518a compared to the plasmid-free *E. coli*. These results and the new MIC and MPC data (MIC in table 1, and MPC in figure 5) we added lines 436-503 showed that sub-MICs of aminoglycosides (antibiotic 1) potentiate the survival of *E. coli* (when carrying the *qnrD*-plasmid) isolates that would further develop QRDR mutations leading to higher fluoroquinolone (antibiotic 2) resistance (see also answer to comment #4).

2. The qnrD plasmid containing ORFs 3 and 4 is not commonly found in *E. coli* (Table S3 and Figure S1). It is adequate that most of the work has been performed in this genetically tractable organism. Nevertheless, it would be important to assess whether the same AG-mediated SOS induction occurs in bacteria that usually carry this type of plasmid (P. mirabilis or other Morganellaceae). This is a simple qRT PCR experiment which could add a lot to the significance of the paper. Perhaps the nitrosative stress (which is obviously a deleterious effect) is particular to E. coli and that could be one of the reasons why this particular plasmid architecture occur rarely (or never?) in this species.

We agree that such a strategy is important for the significance of the paper. The experiments of qRT-PCR quantifying the *sulA* expression level in *Providencia rettgeri* carrying the pDIJ09-518a were performed. They have been included in the revised manuscript (lines 188-191) and shown in the figure S3. Our results showed that the SOS response induction is not increased in *P. rettgeri*/pDIJ09-518a upon sub-MIC of tobramycin, but induced in mitomycin. This confirm that the SOS response is functional in this strain and that AG-mediated SOS induction is specific for *E. coli.*

3. There is no direct evidence provided for increase in 8-oxo-G incorporation. Thus, discussion should carefully be worded (Line 425). Indeed, the target(s) (molecules) with which aminoglycoside interact with as inducers remain unknown, although the aminoglycoside effect is dependent on the GO repair system and indirectly results in DNA damage (Figure 3). It is difficult to extend this to the description that aminoglycosides and others form 8-oxo-G (in L280-282). If any, the aminoglycoside effect is indirect. Clarity is needed.

We agree that we did not evidence the increase of 8-oxo-G incorporation. As suggested we reworded the discussion (524-528) by stating that although we did not identify which target(s) aminoglycosides interact with, guanine oxidation upon aminoglycosides and nitrosative stress exposure has been reported in the literature (doi: 10.1128/jb.183.1.131-138.2001 and doi: 10.1126/science.1219192), leading us to propose our mechanism.

4. Further to the above, the qnrD expression was induced by aminoglycosides (L130). However, it remains unknown whether the elevated 2.8-fold increase in qnrD expression in the presence of tobramycin could produce an impact on fluoroquinolone resistance phenotype. First, an aminoglycoside at 1% MIC is expected to exhibit no antibacterial activity. Thus, a simple fluoroquinolone MIC testing can be done in the presence and the absence of very low levels of aminoglycosides that induce SOS response. A reduced fluoroquinolone susceptibility would argue in favor of author-suggested significance of SOS induction findings.

To investigate the influence of the aminoglycoside-mediated SOS induction of *qnrD* on the level of fluoroquinolone resistance, the minimal inhibitory concentrations (MIC) of nalidixic acid, ciprofloxacin, ofloxacin and levofloxacin have been determined for *E. coli* and *E. coli*/pDIJ09-518a exposed to sub-MICs of ciprofloxacin or tobramycin (Table 1). These results showed that the aminoglycoside-induced SOS response increased fluoroquinolone MIC in line with the increased expression of *qnrD* in *E. coli*. Moreover, to test the possibility that the tobramycin could allow the survival of *E. coli* harbouring *qnrD* plasmids, that further develop stepwise accumulation of QRDR mutations, we determined the mutant prevention concentrations (MPC) of ciprofloxacin, levofloxacin and ofloxacin on a multi-susceptible clinical strain of *E. coli*, ATCC 25922 (reference multi-susceptible strain for antibiotic susceptibility testing), and its derivative harbouring pDIJ09-518a or pDIJ09-518aΔ*qnrD*. We added these findings in the revised manuscript (436-503) as well as in the figure 5. Altogether, our findings showed that aminoglycosides increased *qnrD* gene expression and promote high-level fluoroquinolone resistance.

5. As indicated by authors (L414-415), the functional roles of ORF3 and ORF4 via biochemical studies were not directly measured. Instead, sfiA expression and fluorescence indicators were assessed. This represents a weakness of the study. Can the authors address this?

We agree with that demonstrating the functional roles of ORF3 and ORF4 via biochemical studies is an important point that would have strengthened our study. However, as we mentioned initially, such experimental effort does not invalidate our results and does not reduce its significance. Considering all the reviewer’s recommendations, we choose the other experiments to be considered as top priority (for instance, qRT-PCR on *P. rettgeri*, MPC, MIC, spontaneous mutations). As a follow-up to this initial report, we are currently working on biochemically and genetically characterizing the activities of purified proteins.

6. How do the experiments in figure S2 address plasmid stability during growth by measuring OD? Plasmid stability in the experimental conditions is an important issue.

Initially, experiments were performed in order to assess the stability of pDIJ09-518a carriage in the absence of a fluoroquinolone selective pressure. The schematical figure of our experiment is included in the revised manuscript and showed in figure S2B. We performed colony PCR every 5 days on 5 different colonies, until day 30. The plasmid was still present in the *E. coli* WT strain (figure S2C). These results are mentioned lines 146-153.

7. Comparing Figures 3B and 1E, it can be observed that deletion of both recF and recB have significant effects on SOS induction by AG in cells carrying the qnrD plasmid. These results are very important for the model and a qRT measurement of sulA in recF and recB mutants would be an important confirmation for the results obtained with plasmid based assays, which could be influenced by the reporter plasmid copy number and stability in the different genotypes analyzed.

This approach based on the expression of another SOS regulon gene (*recN*) used to measure the SOS response induction have already been used widely in many studies (e.g. doi: 10.1371/journal.pgen.1001165, doi: 10.1371/journal.pgen.1003421, doi: 10.1093/nar/gkt1259). No influence of the reporter plasmid nor its stability were evidenced. The normalization of the method/apparel is made on the level of fluorescence of the WT strain in the MH medium without antibiotic. To avoid any confusion a sentence referencing this approach has been added to the manuscript (line 651).

8. The complementation of ORF 3 and 4 mutants (Figures 4A and 4C) is incomplete, the same SOS induction level as observed in the wt is never reached. Could this be due to the deletion of one ORF influencing on the expression of the other? Additionally, a more appropriate comparison with cells carrying the empty expression vector may help to solve this issue.

As recommended by the reviewers we modified the figure 4A and showed on the same figure the results for the strains carrying the empty expression vector. In the revised figure 4, we can see for the *sulA* expression:

– ORF3 mutant: the same level in WT with tobramycin

– ORF3 mutant: statistically significant in WT/pDIJ09-518a with tobramycin

– ORF3 complemented mutant: statistically significant in WT with tobramycin (same as WT/pDIJ09-518a *vs* WT with tobramycin).

– ORF4 mutant and complemented: same level quantified and difference statistically significant compared to WT with tobramycin. In these strains the ORF3 expression plays a role.

– ORF3ORF4 double mutant: points out that the plasmid backbone and tobramycin are both necessary for SOS response induction in *E. coli*, since we quantified the same level.

9. In contrast with the detailed analysis described in the initial sections, the information provided on the products of ORF3 and ORF4 is scanty. This is unfortunate as these products and their activities are the main findings of the story. Hence, more detailed characterization of these putative proteins seems advisable to support the authors's model. Bioinformatic analysis should be shown in more detail, the existence of the predicted proteins should be proven and the putative DNA binding activity of the ORF4 product should be shown. Otherwise, the possibility of indirect effect will remain open.

The main result of this work is the novelty of an increased MIC to one antibotics after exposure to another class of antibiotics SOS response, more than the full characterization of unknown ORFs. However, as requested by the reviewer, we added a more detailed Bioinformatics analysis of the products ORF3 and ORF4. In the revised manuscript, lines 385-392, we have included the name of the homologous protein as well as identification accession number for different NCBI data bases that support the existence of these proteins. In Author response image 1 you will see the MSA view of the Blastp/Blastpsi alignment based on the conservation of the amino acids between the query (ORF3 or ORF4) and the subject we have found.

**Author response image 1. sa2fig1:** 

Analysis of Author response image 1 shows how we came to present our results lines 385-392.

Other Comments that should be addressed:1. The study introduces two groups of qnrD plasmids (L77). Do the authors have information about their copy numbers? For example, what is the copy number of pDIJ09-518a? This information will assist in understanding the basal expression of qnrD and the other genes in the absence of an inducer, and thus the significance of aminoglycoside induction of SOS response.

We carried out the plasmid copy number determination by absolute quantification in *E. coli* and *P. rettgeri* and found 230 copies in *E. coli* and 35 in *P. rettgeri*. Since we showed that the *qnrD*-plasmid alone does not induce the SOS response in *E. coli* (figure 1D), we have decided not to add these new results in the revised manuscript to focus on the main message.

2. BLAST was carried out to find homologs of ORF3 and ORF4 (L366-373). Please include gene/product access information for an Erythrobacter spp protein and an O2-responsive regulator from Pedobacter panaciterrae. Are there any reference(s) to support these proteins that are used for comparison?

See point 9 in the section ‘’main concerns’’. Done for the revised manuscript lines 385-392.

3. L326. In the absence of RecB, the SOS response was somehow (1.4-fold change; but seems statistically significant [Figure 3B]) observed in the presence of tobramycin, suggesting the possible presence of a RecF-independent SOS response pathway. For comparison, the control for both recF and recB, the ratios are quite below 1, Hence, the actual changes are larger. Suggest additional discussion. (Additionally, "LB" is indicated in text while Figure 3B shows "MH" – please clarify.)

Sorry for the typo. Results presented in figure 1E have been obtained from experiments performed in MH. We have modified this mistake lines 171 and 328. In order to avoid any misunderstanding, we would like to clarify that in figure 3B we showed a 1.4- fold change in the SOS response induction for the mutant Δ*recF* and not for Δ*recB*. For this latter, we did not find any induction of the SOS*.*

4. L65. Among 53 fully sequenced qnrD plasmids, do they share the same orf3 and orf4 reported in this manuscript? In other word, how often are these ORFs observed in qnrD plasmids?

The ORF3 and ORF4 are only found into the pDIJ09-518a type plasmids. These two ORFs are not present into the bigger-*qnrD* plasmid. We have included one sentence in the revised manuscript lines 77-78 as well as two columns showing the presence (+) of ORF3 and ORF4 in the Table S1.

5. L221. Is the description "However, the underlying mechanism explaining these findings has yet to be identified" simply due to QnrD protection of ciprofloxacin target?

We agree that these findings may be due to the simple protection by *qnrD* of FQ target, but to our knowledge there is no evidence whether it could be due to DNA topology issue or mutation rate.

6. L402-403. The description "We found that ORF4 did not alleviate the SOS response after exposure to aminoglycoside sub-MICs as measured by sfiA expression (Figure 4C)" is not reflective of Figure 4C because orf3 deletion reduced sifA expression (statistically significant per Figure 4C legend). Please verify. As well, obviously, the complementation with pORF4 did not work in Figure 4C.

Our results show that ORF4 did not alleviate the SOS response as our statistical test comparing the ORF4 mutant in tobramycin *vs* the WT in tobramycin is significant. This observation is not observed when we compared the ORF4 mutant in tobramycin *vs* the pDIJ09-518a in tobramycin. It is thus clear that ORF3 is still expressed in the ORF4 mutant as we observed a statistically significance between WT/pDIJ09-518a in tobramycin *vs* the double mutant, being statistically significant.

7. L431. The proposed Model (should be Figure 5, not Figure 4) is considered to reflect the findings. However, the wording should be careful. For example, in L436, the description "the level of oxidized guanine increases" may be revised as "the level of oxidized guanine may increase" as no measurement for 8-oxo-G was done in this study.

Done in the revised manuscript line 533.

8. In contrast with the detailed analysis described in the initial sections, the information provided on the products of ORF3 and ORF4 is scanty. This is unfortunate as these products and their activities are the main findings of the story. Hence, more detailed characterization of these putative proteins seems advisable to support the authors's model. Bioinformatic analysis should be shown in more detail, the existence of the predicted proteins should be proven and the putative DNA binding activity of the ORF4 product should be shown. Otherwise, the possibility of indirect effect will remain open.

See point 9 in the section ‘’main concerns’’. Done for the revised manuscript lines 385-392.

9. Absolute numbers of β-gal activities should be provided when lac fusions are used. This would help the reader to assess the magnitude of the phenomena described in the manuscript.

We did not perform fusion transcription with lacZ in this study, but we have used the MG1656 strain, in order to perform the further biochemically study on the proteins encoded by ORF3 and ORF4 genes.

10. Please revise the statement that you proved functionality of the SOS box in the qnr locus of the plasmid family under study. You did so in the archetypal plasmid pDIJ109-518a. The writing may however suggest that functionality was proven in the entire plasmid collection provided in Table S3.

We have modified the sentence in the revised manuscript line 124.

Reviewer 2 (Recommendations for the authors):1. Some of the data files (spreadsheets) have French words/phrases. If they are intended to be published alongside the paper, these terms should be translated to English.

Done.

2. Figure S1 and introduction. Are ORFs 3 and 4 from the "small" qnr plasmids studied in this work also present in the "large" qnr plasmid? This is not clear from the text or Figure S1. As an alternative, Table S3 could indicate if ORFs 3 and 4 homologues are present in each of the qnrD plasmids.

Done. See our answer in ‘’others comments’’ point 4 for L65 and present in the revised manuscript.

3. This is a matter of choice, but sulA is a more widely use gene name than sfiA, and should be used throughout the text. sfiA is the original name, but modern gene annotations in bacterial genomes use sulA.

Done all over the revised manuscript.

4. Lines 118-119: Revise this information. In figure 1A, the consensus observed upstream to qnrD actually has 15 out of 16 bases identical to the *E. coli* consensus shown.

Done line 139 of the revised manuscript.

5. Lines 291 to 293. Phrasing is not good, suggest: the SOS response induction could result from the deleterious effects…

Done lines 294-295 of the revised manuscript.

6. Lines 402-403: "We found that ORF4 did not alleviate the SOS response after exposure to aminoglycoside sub-MICs, as measured by sfiA expression". In my view the SOS response is attenuated, although not completely, after ORF4 deletion. Maybe a more precise statement would be: "We found that ORF4 deletion does not completely abolish SOS induction by aminoglycoside sub-MICs, as measured by sfiA expression".

Done line 420 of the revised manuscript.

7. There is an unnecessary repetition of results in Figures 4A, 4C and 4E (wt, wt/pDIJ09). All these results could be merged into a single graph.

Done for Figure 4A

8. Data from Figure 4B and the model depicted in Figure 5 implicate that ORF4 can act on its own (independently from ORF3 or stressors) as a negative regulator of hmp, contributing to nitrosative stress. Nevertheless, if this was the case, I would expect the following in qRT-PCR experiments shown in Figure 4D:Wt/pDIJ09 – lower hmp levels than cells without plasmid, due to (ORF4) action.Wt/pDIJ09 del ORF4 – same hmp levels as cells without plasmid. No negative regulator present in spite of the rest of plasmid.Wt/pDIJ09 del ORF4/pORF4 – lower hmp levels than cells without plasmid, due to in trans negative regulator (ORF4) action.Can the authors perhaps add some comment? Is it possible that there is another interpretation for these data?

We tried to simplify and summarize the mechanism in the scheme of the figure 6. But, we agree that we did not evidence that ORF4 acts independently from ORF3 or stressors.

– The WT represents the baseline level of the expression of hmp needed for the regular detoxification of NO.

– In WT/pDIJ09-518a, hmp could have been expected to be less expressed because of the presence of ORF4. But, ORF3 is present in the plasmid backbone and thus increases NO (See figure 4D with pDIJ09-518a in red > WT in white). Therefore, ORF4 inhibits a higher rate of hmp expression than at the baseline. This compensation may explain that hmp levels are not lower than those in cells without plasmid.

– In ΔORF4, there is no inhibition of hmp by ORF4, and hmp is stimulated by NO generated by ORF3.

– In ΔORF4/pORF4 (orange histogram), over-expression of ORF4 can inhibit the hmp expression despite the presence of ORF3.

9. Lines 622-623 mention a mutation rate analysis, but such experiments are not shown in the paper.

Modified in the revised manuscript and shown in the Figure S5.

10. Viability after tobramycin treatment (lines 601-606): It is not clear for how long cells were exposed to the antimicrobial.

Modified in the revised manuscript (line 705).

11. Material and Methods or Figure legends should be more clear regarding the concentration of antimicrobials used in each experiment. For example, when comparing *E. coli* cells carrying or not the qnrD plasmid, each strain was treated with different sub-MIC concentrations of cipro to account for their different resistance levels? In Materials and methods (lines 479-481), 1% of the MIC of Cipro (0.06 µg/mL) is indicated as the sub-MIC concentration used. However, Table S1 indicates a MIC of 0.004 µg/mL for the wt strain, and 0.094 µg/mL for the strain carrying the qnr plasmid. This information is confusing and should be revised.

We have performed MIC testing E-tests. Conversely to the microdilution broth method, it is not a twofold dilutions test. Then results can be a little bit different. Pinpointing 0.094, the microdilution method would have provided results such as 0.125, 0.06 or 0.03 μg/ml. We double checked the MIC of *E. coli* WT and *E. coli*/pDIJ09-518a using the microdilution broth method (Sensititre broth microdilution, Thermo Scientific). For the MIC value 0.094 μg/ml, as expected, we found we found 0.06. Given this result, and that we performed previously all the MICs using E-test, we kept the E-test results (moreover well accepted in clinical microbiology laboratories for diagnosis).

12. Were experiments measuring SOS induction with the GFP reporter performed in LB or MH medium? Material and Methods (lines 559-561) is contradictory on this regard. Figure legends indicate LB, but the Y axis of figures 1E and 3B show MH.

Modified in the revised manuscript.

13. Line 588 mention levofloxacin and ofloxacin, but these antimicrobials were not used in this work.

It was a mistake. Since we have added new data with these antimicrobials, as recommended by reviewers, we kept the paragraph lines 680-683.

Reviewer 3 (Recommendations for the authors):1. The authors use the term "aminoglycosides" in multiple sections of the manuscript, including the title, the abstract and the discussion. However, only one aminoglycoside (tobramycin) has been used throughout the study. Gentamicin was used at the beginning of the study but disappears at some point. Please clarify.

First, we demonstrated the SOS induction upon tobramycin and gentamicin treatment (figures 1C, 1D and 1F). Once this induction demonstrated, we aimed at deciphering the mechanism. For that purpose, we decided to use only a single aminoglycoside since both had been evidenced to induce the SOS.

[Editors' note: further revisions were suggested prior to acceptance, as described below.]

Essential revisions:Whilst reviewers concurred that your manuscript has improved, significant concerns around complementation still remain. Please address these. Other points are indicated below.1. The manuscript is somewhat confusing regarding Figure and supplemental figure names. It is often difficult to understand which figured is being commented in the text (Examples: line 228, line 283, line 372). Also, supplemental figures are not numbered in the final document, making it hard to follow the text.

We agree with the reviewer that the supplemental figures were not numbered according to the modifications requested by the editorial office. We modified them. In addition, the former figure 1—figure supplement 1 with the pie-charts showing the distribution of *qnrD*-plasmids among *Enterobacterales* and *Morganellaceae* did not really match with the former figure 1. We guess it led to be somewhat confusing to read appropriately the manuscript. To avoid such a confusion, this figure has been added in the manuscript since it does not match with the former figure 1.

2. Page 8 lines 187-191. This result deserves a better discussion in the paper. The burden presented by this plasmid for *E. coli* is not present in Providencia. Therefore, this may one of the reasons why qnrD plasmids are rarely found in *E. coli*. SOS induction per se is a symptom of a stressed cell in a disadvantageous situation. Therefore, lack of SOS induction for Providencia means that this species is not harmed by presence of the plasmid, whereas the opposite is true for E. coli.

We agree with the reviewer that the lack of SOS induction in *Providencia* spp. upon aminoglycosides exposure turns to be due to the lack of burden caused by the *qnrD*-plasmid and observed in *E. coli* with tobramycin. *qnrD*-plasmids are mainly described in *Proteeae..* In *E. coli* carrying the *qnrD*-plasmid, as well as in *Providencia* spp., no induction of the SOS response was found without exposing the cells to tobramycin. We agree that we can speculate that such a *qnrD*-plasmid is harmless for *Providencia* spp. conversely to *E. coli* and that may explain why these plasmids are rarely found in *E. coli*.

We have conducted a protein sequence alignment of D4C4X8_PRORE (UniProt annotation of *Providencia rettgeri* Hmp *protein*) using BLASTp with the protein Hmp from *E. coli*. The results of the alignment, in term of length of sequence showed a 63,38% protein identity with *E. coli* str. K-12 substr MG1655 (sequence identity NP_4170471.1). When analyzing deeper the *Providencia rettgeri* Hmp protein sequence using the MSA Viewer, we found differences between the two species in term of amino acids. This result has been added in the manuscript (lines 193-200) and in the figure 2—figure supplement 2 (panel B)

We did not find any references about the *Proteeae hmp* gene characterization in literature. To follow the lead provided by the reviewer in order to explain that the burden presented by the *qnrD*-plasmid in *E. coli* is not present in *Providencia* spp., we can consider two hypothesis: (i) Hmp is not inhibited by ORF4 in *Providencia* spp. due to only 63.4% of identity with *E. coli* Hmp, allowing then Hmp to play its role in NO detoxification and (ii) ORF3 is not active in *Providencia* spp. limiting the amount of NO production. This part has been added in the discussion lines 546-549.

3. The plasmid used for complementation in experiments shown in Figure 4A clearly affects SOS induction by Tobramycin in cells carrying pDIJ09 (second versus fourth brown bar in figure 4A). This complicates the interpretation of all complementation experiments and should be revisited and the discussion should carefully reflect this limitation.

We agree with the reviewer that there is a mistake on the figure 4 of the last revised manuscript (round 1). Last time, we modified the figure 4A as recommended by the reviewer. However, in order to generate the tobramycin results of figure 4A, we unfortunately made a mistake by copying-pasting the LB raw data for WT/pΦ and WT/pDIJ09-518A/pΦ (grey and black bar) from the former figure S4A rather than tobramycin raw data for WT/pDIJ09-518A/pΦ from S4B. This led to wrongly show that pΦ affects the SOS induction in cells carrying pDIJ09-518a. In the former figure S4, we had clearly shown that the *sulA* expression is increased in *E. coli*/pDIJ09-518a upon tobramycin exposure conversely to antibiotic-free LB wherein no induction of SOS was noticed for WT/pΦ and WT/pDIJ09-518a/pΦ. The raw data were available since the first submission to confirm we just made a mistake by copying-pasting data to generate the figure 4 in the last revised manuscript.

Taking account into this mistake, we modified the figure 4 (now figure 5 in the new revised manuscript). The pΦ empty vector co-carried with pDIJ09-518a seems to decrease the *sulA* expression without tobramycin treatment (grey bar, Figure 4—figure supplement 1). We did find any explanation for this observation but it may explain why the sulA expression upon tobramycin exposure is a bit lower for WT/pDIJ09-518a/pΦ than for WT/pDIJ09-518a (second and third brown bar of the new figure 5). Either way, such a result cannot call into question our results since we still observed an increase of *sulA* expression.

There is no reference for the pTOPO vector used for complementation. Does it carry kanamycin or another aminoglycoside resistance? Could some sort of low-level cross resistance conferred by the plasmid be interfering with the effects of Tobramycin? Perhaps complementation with a chromosomal integration of ORFs 3 and 4 as done in other parts of the paper would be better.

In figure 4 (formerly Figure 3) and figure 5 (formerly Figure 4) as well as in the table in supplementary file 3, pΦ stands for the pCR2.1. (from ThermoFisher Scientific) empty vector that we unfortunately called pTOPO in the material and methods. It has been modified in the revised manuscript (line 613). pCR2.1. carries a kanamycin resistance determinant. However, (1) this plasmid was not used to perform any MIC determination relative to level of fluoroquinolone resistance and (2) we showed that pΦ does not increase *sulA* expression not only in the wild-type *E. coli* strain but also the wild-type *E. coli* strain carrying pDIJ09-518a, without tobramycin (figure 4—figure supplement 1A). All together, we think that these experimental controls ruled out any interfering effect from the kanamycin resistance determinant encoded within pCR2.1. (pΦ).

4. In figure 3D, why does deletion of hmp does not increase sulA induction after Tobra treatment? The result does not make sense when compared to the overexpression and should be discussed in further detail.

It turns out there is a misunderstanding regarding the figure 4D (formerly 3D). In this latter, the SOS response induction was relatively quantified and we showed a statistically significant increase of *sulA* expression in the *hmp* mutant carrying the pDIJ09-518a (Δ*hmp*/pDIJ09-518a) upon tobramycin exposure compared to ATB-free LB (third bar – green – in figure 4D). In the strain carrying the *qnrD*-plasmid pDIJ09-518a (WT/pDIJ09-518a/pHmp), overexpression of hmp (pHmp) abrogated the SOS response induction after tobramycin treatment (figure 4D, second bar). Moreover, after scavenging NO by cPTIO, we observed the same level of *sulA* expression for all the isogenic strains (white bars, 4^th^ to 6^th^ bar). As mentioned (lines 360-375) and discussed (531-545) lines in the manuscript, this pointed out that, in the strain carrying pDIJ09-518a in the presence of tobramycin, NO is accumulating and the NO-detoxifier Hmp is not functional.

5. Lines 482 to 490 are very confusing and need re-writing. The MPC experiments need a detailed description in the methods section.

In the revised manuscript, line 498, we have clearly mentioned that QDRD mutations have been assessed in *E. coli* ATCC 25922*/*pDIJ09-518a isolates harvested from the MPC assay. We think it will be less confusing for the reader when comparing the text with the table 2.

6. Mutagenesis data should ideally represent at least 3 experiments. Authors should also describe in more detail how these experiments were performed (initial inoculum, hours of growth in the presence of the drug, etc).

Lines 728-732, we have described how the experiments were performed: initial inoculum was made from single colony (total of 12 colonies for each strain: 6 colonies for each strain in 2 independent assays) and grown over night in LB in the presence or the absence of sub-minimum inhibitory concentration of tobramycin.

7. The titles of Tables 1 and 2 includes "fluoroquinolone", while nalidixic acid is not a fluoroquinolone. A footnote could be helpful. In Abstract, please write "encodes", not "encode" (L47) and write "codes", not "code" (L48).

We have done all the modifications all over the revised manuscript.

8. As ORF3 and ORF4 have not been characterized, it should be made clear that the identities of the ORF3 and ORF4 products are putative.

We have done all the modifications all over the revised manuscript.

Reviewer #2 (Recommendations for the authors):This version of the manuscript has addressed many of the previous points raised by the reviewers. The work is interesting, but there are still some issues that need to be addressed:For some reason, this version of the paper is very confusing regarding Figure and supplemental figure names. It is often difficult to understand which figured is being commented in the text (Examples: line 228, line 283, line 372). Also, supplemental figures are not numbered in the final document, making it hard to follow the text.

We agree with the reviewer that the supplemental figures were not numbered according to the modifications requested by the editorial office. We modified them. In addition, the former figure 1—figure supplement 1 with the pie-charts showing the distribution of *qnrD*-plasmids among *Enterobacterales* and *Morganellaceae* did not really match with the former figure 1. We guess it led to be somewhat confusing to read appropriately the manuscript. To avoid such a confusion, this figure has been added in the manuscript since it does not match with the former figure 1.

Page 8 lines 187-191. In my opinion, this result deserves a better discussion in the paper. The burden presented by this plasmid for *E. coli* is not present in Providencia. Therefore, this may one of the reasons why qnrD plasmids are rarely found in *E. coli*. SOS induction per se is a symptom of a stressed cell in a disadvantageous situation. Therefore, lack of SOS induction for Providencia means that this species is not harmed by presence of the plasmid, whereas the opposite is true for *E. coli*.

We agree with the reviewer that the lack of SOS induction in *Providencia* spp. upon aminoglycosides exposure turns to be due to the lack of burden caused by the *qnrD*-plasmid and observed in *E. coli* with tobramycin. *qnrD*-plasmids are mainly described in *Proteeae..* In *E. coli* carrying the *qnrD*-plasmid, as well as in *Providencia* spp., no induction of the SOS response was found without exposing the cells to tobramycin. We agree that we can speculate that such a *qnrD*-plasmid is harmless for *Providencia* spp. conversely to *E. coli* and that may explain why these plasmids are rarely found in *E. coli*.

We have conducted a protein sequence alignment of D4C4X8_PRORE (UniProt annotation of *Providencia rettgeri* Hmp *protein*) using BLASTp with the protein Hmp from *E. coli*. The results of the alignment, in term of length of sequence showed a 63,38% protein identity with *E. coli* str. K-12 substr MG1655 (sequence identity NP_4170471.1). When analyzing deeper the *Providencia rettgeri* Hmp protein sequence using the MSA Viewer, we found differences between the two species in term of amino acids. This result has been added in the manuscript (lines 193-200) and in the figure 2—figure supplement 2 (panel B)

We did not find any references about the *Proteeae hmp* gene characterization in literature. To follow the lead provided by the reviewer in order to explain that the burden presented by the *qnrD*-plasmid in *E. coli* is not present in *Providencia* spp., we can consider two hypothesis: (i) Hmp is not inhibited by ORF4 in *Providencia* spp. due to only 63.4% of identity with *E. coli* Hmp, allowing then Hmp to play its role in NO detoxification and (ii) ORF3 is not active in *Providencia* spp. limiting the amount of NO production. This part has been added in the discussion lines 546-549.

The plasmid used for complementation in experiments shown in Figure 4A clearly affects SOS induction by Tobramycin in cells carrying pDIJ09 (second versus fourth brown bar in figure 4A). This complicates the interpretation of all complementation experiments and should be revisited. There is no reference for the pTOPO vector used for complementation. Does it carry kanamycin or another aminoglycoside resistance? Could some sort of low-level cross resistance conferred by the plasmid be interfering with the effects of Tobramycin? Perhaps complementation with a chromosomal integration of ORFs 3 and 4 as done in other parts of the paper would be better.

We agree with the reviewer that there is a mistake on the figure 4 of the last revised manuscript (round 1). Last time, we modified the figure 4A as recommended by the reviewer. However, in order to generate the tobramycin results of figure 4A, we unfortunately made a mistake by copying-pasting the LB raw data for WT/pΦ and WT/pDIJ09-518A/pΦ (grey and black bar) from the former figure S4A rather than tobramycin raw data for WT/pDIJ09-518A/pΦ from S4B. This led to wrongly show that pΦ affects the SOS induction in cells carrying pDIJ09-518a. In the former figure S4, we had clearly shown that the *sulA* expression is increased in *E. coli*/pDIJ09-518a upon tobramycin exposure conversely to antibiotic-free LB wherein no induction of SOS was noticed for WT/pΦ and WT/pDIJ09-518a/pΦ. The raw data were available since the first submission to confirm we just made a mistake by copying-pasting data to generate the figure 4 in the last revised manuscript.

Taking account into this mistake, we modified the figure 4 (now figure 5 in the new revised manuscript). The pΦ empty vector co-carried with pDIJ09-518a seems to decrease the *sulA* expression without tobramycin treatment (grey bar, Figure 4—figure supplement 1). We did find any explanation for this observation but it may explain why the sulA expression upon tobramycin exposure is a bit lower for WT/pDIJ09-518a/pΦ than for WT/pDIJ09-518a (second and third brown bar of the new figure 5). Either way, such a result cannot call into question our results since we still observed an increase of *sulA* expression.

In figure 4 (formerly Figure 3) and figure 5 (formerly Figure 4) as well as in the table in supplementary file 3, pΦ stands for the pCR2.1. (from ThermoFisher Scientific) empty vector that we unfortunately called pTOPO in the material and methods. It has been modified in the revised manuscript (lines 604-605). pCR2.1. carries a kanamycin resistance determinant. However, (1) this plasmid was not used to perform any MIC determination relative to level of fluoroquinolone resistance and (2) we showed that pΦ does not increase *sulA* expression not only in the wild-type *E. coli* strain but also the wild-type *E. coli* strain carrying pDIJ09-518a, without tobramycin (figure 4—figure supplement 1A). All together, we think that these experimental controls ruled out any interfering effect from the kanamycin resistance determinant encoded within pCR2.1. (pΦ).

In figure 3D, why deletion of hmp does not increase sulA induction after Tobra treatment? The result does not make sense when compared to the overexpression and should be better discussed.

It turns out there is a misunderstanding regarding the figure 4D (formerly 3D). In this latter, the SOS response induction was relatively quantified and we showed a statistically significant increase of *sulA* expression in the *hmp* mutant carrying the pDIJ09-518a (Δ*hmp*/pDIJ09-518a) upon tobramycin exposure compared to ATB-free LB (third bar – green – in figure 4D). In the strain carrying the *qnrD*-plasmid pDIJ09-518a (WT/pDIJ09-518a/pHmp), overexpression of hmp (pHmp) abrogated the SOS response induction after tobramycin treatment (figure 4D, second bar). Moreover, after scavenging NO by cPTIO, we observed the same level of *sulA* expression for all the isogenic strains (white bars, 4^th^ to 6^th^ bar). As mentioned (lines 360-375) and discussed (531-545) lines in the manuscript, this pointed out that in the strain carrying pDIJ09-518a in the presence of tobramycin, NO is accumulating and the NO-detoxifier Hmp is not functional.

Lines 482 to 490 are very confusing and need re-writing. The MPC experiments need a detailed description in the methods section.

In the revised manuscript, line 498, we have clearly mentioned that QDRD mutations have been assessed in *E. coli* ATCC 25922*/*pDIJ09-518a isolates harvested from the MPC assay. We think it will be less confusing for the reader when comparing the text with the table 2.

Mutagenesis data should ideally represent at least 3 experiments. Authors should also describe in more detail how these experiments were performed (initial inoculum, hours of growth in the presence of the drug, etc).

Lines 728- 732, we have described how the experiments were performed: initial inoculum was made from single colony (total of 12 colonies for each strain: 6 colonies for each strain in 2 independent assays) and grown over night in LB in the presence or the absence of sub-minimum inhibitory concentration of tobramycin.

Reviewer #3 (Recommendations for the authors):The authors have done a good job responding to the reviewers' suggestions. Relevant points have been clarified and risky statements have been toned down in the revised version. The story remains incomplete as ORF3 and ORF4 have not been characterized. As a consequence, a crucial point in the story (the mechanism that triggers NO accumulation) remains enigmatic. Despite this shortage, I agree with the authors that qnr-mediated induction of the SOS response by aminoglycosides is an interesting phenomenon. My only suggestion at this point is to modify the text wherever pertinent (e. g., lines 47-49 and 388-396) to make it clear that the identities of the ORF3 and ORF4 products are putative.

We have done all the modifications all over the revised manuscript.